# Multivalent binding of the tardigrade Dsup protein to chromatin promotes yeast survival and longevity upon exposure to oxidative damage

Rhiannon R. Aguilar [1,2,7], Laiba F. Khan [3,7], Christopher K. Cummins[4], Nina Arslanovic[1], Thea Grauer[1], Kaylah Birmingham[1,5], Kritika Kasliwal[1,6], Spike D. L. Posnikoff[1], Ujani Chakraborty[1], Allison R. Hickman [3], Rachel Watson[3], Ryan J. Ezell[3], Sabrina R. Hunt[3], Laylo Mukhsinova [3], Hannah E. Willis [3], Martis W. Cowles[3], Richard Garner[1,6], Abraham Shim [1,6], J. Ignacio Gutierrez [1], Bryan J. Venters [3], Matthew R. Marunde[3], Brian D. Strahl [4], Michael-Christopher Keogh [3,8] ✉ & Jessica K. Tyler [1,8] ✉

Tardigrades are remarkable in their ability to survive extreme environments. The damage suppressor (Dsup) protein is thought to contribute to their extreme resistance to reactive oxygen species (ROS) generated by irradiation. Here we show that expression of *Ramazzottius varieornatus* Dsup in *Saccharomyces cerevisiae* reduces oxidative DNA damage and extends lifespan in response to chronic oxidative genotoxicity. Dsup uses multiple modes of engagement with the nucleosomal H2A/H2B acidic patch, H3/H4 histone tails and DNA to bind across the yeast genome without bias. Effective chromatin binding and genome protection requires the Dsup HMGN-like motif and C-terminal sequences. These findings give precedent and mechanistic understanding for engineering an organism by physically shielding its genome to promote survival and longevity in the face of oxidative damage.

Tardigrades (also termed water bears) are an invertebrate phylum of > 1200 species with broad-reaching habitats. Many can survive desiccation, extreme temperatures, high pressure, intense irradiation, and exposure to space[1]. The mechanisms by which various tardigrade species resist such extreme stressors are poorly understood. *Ramazzottius varieornatus* is highly resistant to ionizing radiation (IR); capable of surviving > 48 h after a dose of 4000 Gy[2], compared to the human $LD_{50}$ of ~4.5 Gy[3]. The *R.varieornatus* Dsup (Damage suppressor) protein is chromatin associated and predicted to promote IR resistance, being absent from IR sensitive tardigrade species[4]. Indeed, when expressed in human cells, Dsup localizes to nuclear DNA and confers IR-resistance accompanied by reduced levels of DNA single- and double-strand breaks (SSBs and DSBs)[4]. Radioprotection is also conferred when Dsup is expressed in tobacco plants[5], flies[6] and mice[7].

While IR can directly induce SSBs and DSBs, much of its genotoxicity is mediated by hydroxyl radicals (OH·) generated when

[1]Department of Pathology and Laboratory Medicine, Weill Cornell Medicine, New York, NY, USA. [2]Weill Cornell / Rockefeller / Sloan-Kettering Tri-Institutional MD-PhD Program, New York, NY, USA. [3]EpiCypher Inc., Durham, NC, USA. [4]Department of Biochemistry and Biophysics and Lineberger Comprehensive Cancer Center, University of North Carolina at Chapel Hill, Chapel Hill, NC, USA. [5]Pharmacology Graduate Program, Weill Cornell Medicine, New York, NY, USA. [6]Biochemistry, Cellular, and Molecular Biology Graduate Program, Weill Cornell Medicine, New York, NY, USA. [7]These authors contributed equally: Rhiannon R. Aguilar, Laiba F. Khan. [8]These authors jointly supervised this work: Michael-Christopher Keogh, Jessica K. Tyler. ✉e-mail: mkeogh@epicypher.com; jet2021@med.cornell.edu

radiation interacts with water molecules[8]. Indeed high-energy OH[·] are the most powerful oxidant among the reactive oxygen species (ROS), reacting with DNA bases to form lesions (including 8-oxoguanine; 8-oxo-G), while oxidation of the deoxyribose backbone dissociates sugar-phosphate bonds leading to DNA breaks[9]. Consistent with Dsup protecting against OH[·], it also reduces the number of DNA breaks in human cells exposed to hydrogen peroxide ($H_2O_2$)[4,10]. Throughout life, oxidative DNA damage is continually generated from aerobic metabolism, with the resulting mutations thought to contribute to the ageing process[11] and the development of age-related diseases[12], such as neurodegeneration[13] and cancer[14,15]. Further, most cancer treatments cause oxidative DNA damage and strand breaks, contributing to long-term side effects in survivors[16]. As such, the means by which proteins such as Dsup protect the genome from oxidative damage are of extreme interest.

*R.varieornatus* Dsup is a 445 amino acid protein[17] that is intrinsically disordered[17–19]. Of note, disorder at their N- and C-termini is a feature of proteins that scan and engage DNA, consistent with a DNA-binding role for Dsup[20,21]. C-terminal deletion (Δ aa 208-445) abrogates Dsup binding to naked DNA or human chromatin[4]. Indeed, Dsup binds with higher affinity to reconstituted chromatin over free DNA, and sequences within aa 360–445 are required for the association with chromatin and protection from OH[·] induced DSBs[22]. While Dsup upregulates the expression of various DNA repair genes in HEK293 cells[10], the protein also prevents DNA damage by binding chromatin in a reconstituted system lacking DNA repair factors[22].

Within the Dsup C-terminal region, an eight amino acid stretch (aa 363–370, RRSSRLTS) has homology to the core consensus (RRSARLSA) of the nucleosome binding region of vertebrate High Mobility Group-N (HMGN) proteins[22–24]. The chromatin binding of HMGN proteins influences a wide variety of functions (including embryogenesis, development and disease protection) across diverse cell types and species[25]. In the prevailing model revealed by the in vitro reconstituted system, Dsup protects the genome from DNA damage by physically shielding chromatin from hydroxyl radicals, in a manner that involves its HMGN-like motif and likely additional C-terminal sequences[22]. However, whether this model operates in vivo is unknown.

Here, we show that when expressed in budding yeast at histone-like stoichiometry *R.varieornatus* Dsup uses its HMGN-like motif and adjacent C-terminal sequences to protect the genome from oxidative DNA damage in a manner dependent on chromatin engagement but independent of scavenging hydroxyl radicals or differentially regulating DNA repair pathways. Dsup expression also extends yeast replicative lifespan in the face of chronic endogenous oxidative DNA damage. A detailed analysis of [Dsup: nucleosome] engagement shows that its HMGN-like motif mediates interaction with the H2A/H2B acidic patch on the nucleosome surface, while its distal C-terminal sequences bind DNA. Of note, such multivalent binding supports the observed broad engagement with in vivo chromatin, independent of the landscape of histone post-translational modifications (PTMs). Our studies indicate that tardigrade Dsup can be introduced to a heterologous in vivo system and confer viability and longevity in the face of elevated levels of oxidative damage. This is achieved by physically coating the chromatinized genome via multivalent interactions to prevent hydroxyl radicals from damaging genomic DNA.

## Results

### Heterologous expression of *R. varieornatus* Dsup in budding yeast protects against oxidative damage and promotes longevity in the face of increased oxidative stress

To initiate this study we expressed epitope tagged 6His-Dsup-FLAG (hereafter Dsup-FLAG or Dsup (WT)) in yeast under the constitutive high output *TDH3* promoter[26] (Supplementary Data File 1), with the goal of achieving in vivo protein levels sufficient to coat the genome. This yielded Dsup-FLAG of similar abundance to H2B-FLAG (Fig. 1a). To investigate the response of Dsup-FLAG yeast to chronic oxidative damage, we performed serial dilution assays on plates containing $H_2O_2$, observing a ~ 25-fold increased survival relative to yeast lacking Dsup (Fig. 1b). This did not extend to general genotoxin protection, since Dsup-FLAG slightly decreased yeast survival in response to non-oxidative DNA-damaging agents such as alkylating methyl methanesulfonate (MMS), DNA-intercalating Zeocin, or ultra violet (UV) light (Fig. 1b).

In reconstituted assays recombinant Dsup protects chromatinized plasmid DNA from DSBs caused by hydroxyl radicals[22], so we asked if Dsup expression protected the in vivo yeast genome from oxidative DNA damage. 8-oxoguanine (8-OHdG) is generated when ROS react with DNA[27], so we quantified this base modification after transient exposure to $H_2O_2$ and observed a significant reduction in the presence of Dsup (Fig. 1c).

ROS and oxidative damage increase with age, and reducing oxidative damage extends the lifespan of multiple species (yeast, worms, fruit flies, mice[28]), while elevated ROS production shortens lifespan[29]. We thus asked if Dsup expression could extend yeast lifespan. In otherwise WT yeast Dsup had a negligible impact on chronological lifespan (the length of time a cell survives in a non-dividing state; Supplementary Fig. 1a), while replicative lifespan (the maximum number of times a cell can divide) was slightly reduced (Supplementary Fig. 1b). Cells lacking the superoxide dismutase (*SOD*) genes are deficient in their ability to process both endogenous and exogenous ROS. As a result, they accumulate oxidative stress and damage, such that yeast lacking Sod1 have a shortened replicative lifespan[30]. When expressed in *sod1Δ* yeast, Dsup significantly increased replicative lifespan (Fig. 1d), suggesting enhanced survival and longevity in the face of chronic oxidative damage.

### Dsup promotes yeast survival upon oxidative damage via its C-terminal sequences, and reduces the cellular redox state via its N-terminal sequences

Vertebrate High Mobility Group-N (HMGN) proteins[22] contain a conserved HMGN motif (core consensus RRSARLSA[31]) required for chromatin binding and protein fuction[25,32] (Fig. 2a). The Dsup C-terminal region contains an eight amino acid stretch with homology to this consensus[22] (the HMGN-like motif: aa363-370, RRSSRLTS: Fig. 2a), suggesting physiological relevance. We thus made mutant forms of Dsup by substituting three positively charged arginines in the motif with negatively charged glutamic acid (Dsup HMGN-3R/3E: R363E/R364E/R367E), or by deleting the entire C-terminus including the HMGN-like motif (Dsup ΔHMGN ΔC: Δ360-445); alleles previously investigated in vitro[22]. Dsup contains a predicted nuclear localization signal (NLS)[33] removed by ΔHMGN ΔC, so to this we added a repeated SV40 NLS (PKKKRKVPKKKRKV)[34] to create Dsup ΔHMGN ΔC + NLS (Fig. 2a, b). By immunofluorescence Dsup (WT), Dsup HMGN-3R/3E and Dsup ΔHMGN ΔC + NLS primarily localized to the nucleus, while Dsup ΔHMGN ΔC was primarily cytoplasmic, presumably due to removal of the predicted NLS (Fig. 2c). Dsup ΔHMGN ΔC was thus omitted from further in vivo study. Importantly, the Dsup HMGN-3R/3E and Dsup ΔHMGN ΔC + NLS proteins were expressed at least as well as Dsup (WT) in yeast (Fig. 2d), and each did not significantly impact cell growth (Fig. 2e).

We next examined the ability of Dsup mutants to enhance yeast survival through chronic or acute oxidative stress; the former by growth on plates containing $H_2O_2$ (Fig. 3a), the latter by exposing cells to $H_2O_2$ in liquid culture for 1.5 hours before recovery on plates with no oxidizing agent. (Fig. 3b). In each case Dsup HMGN-3R/3E protected cells similarly to Dsup (WT), while Dsup ΔHMGN ΔC + NLS

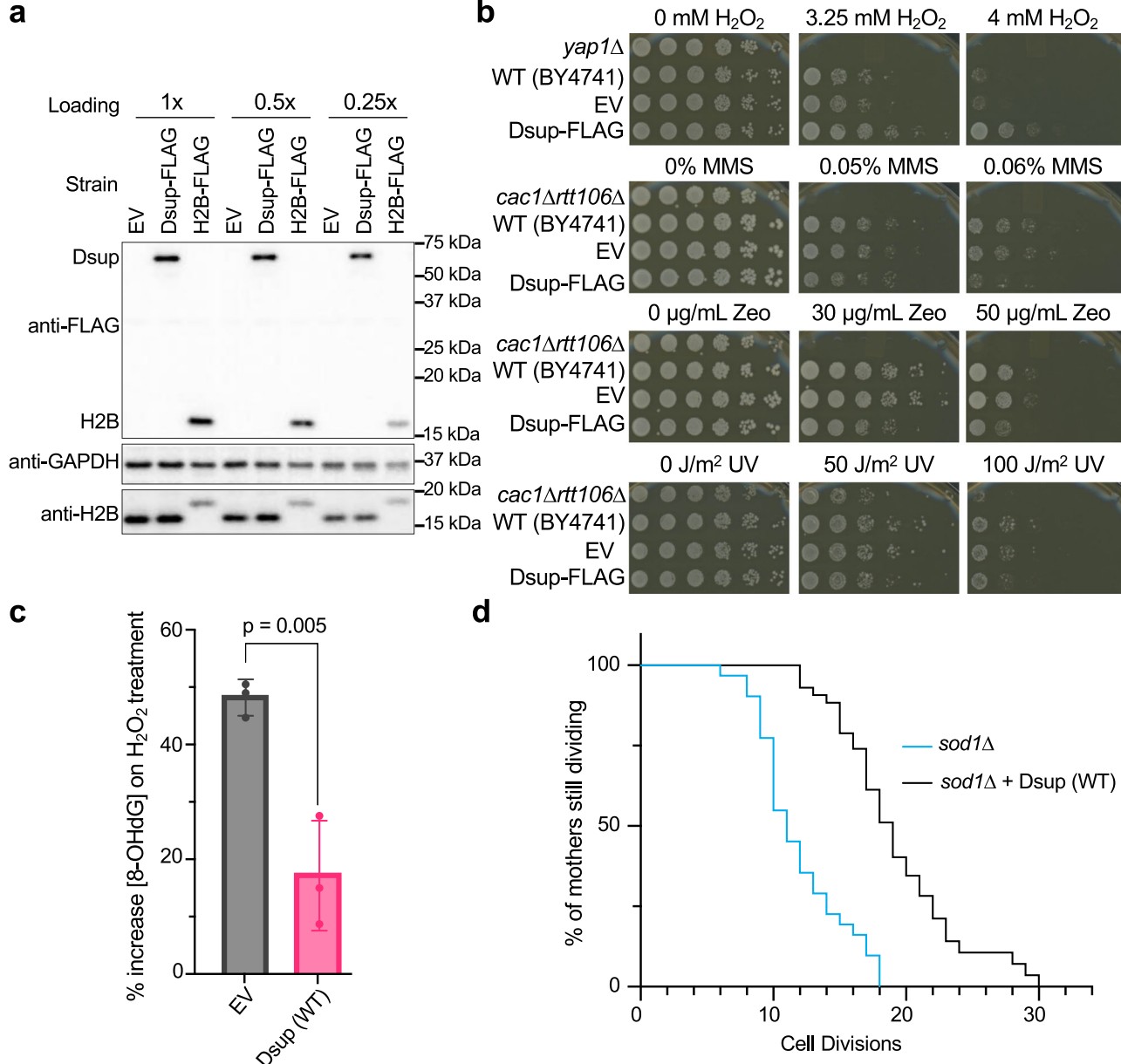

**Fig. 1 | Heterologous expression of tardigrade Dsup in budding yeast promotes survival after chronic exposure to an oxidative DNA damaging agent, reduces related DNA damage, and extends lifespan through chronic endogenous oxidative damage. a** Immunoblot of Dsup-FLAG (from pTDH3-6His-Dsup-FLAG) and H2B-FLAG in yeast strains containing a single integrated copy of each tagged gene (Supplementary Data File 1B). The same protein samples were ran on a second gel for detection of GAPDH and H2B. Three dilutions of protein extracts loaded as indicated. EV, Empty vector. **b** Relative sensitivity of yeast strains (five-fold serial dilutions) to indicated doses of hydrogen peroxide (H$_2$O$_2$), methyl methanesulfonate (MMS), Zeocin (Zeo), or UV. *Yap1Δ* is a positive control for sensitivity to

oxidative DNA damage (H$_2$O$_2$). *Cac1Δrtt106Δ* is a positive control for sensitivity to other genotoxins. WT (BY4741), wild-type yeast. Dsup (WT), Dsup1 (aa 1-455). **c** Yeast cells expressing Dsup (WT) show reduced levels of 8-hydroxy 2 deoxyguanosine (8-OhdG) in response to oxidative DNA damage (120 min in 10 mM H$_2$O$_2$; mean and standard deviation from three independent experiments; *p*-value determined using a two-sided Student's t-test). **d** Yeast cells expressing Dsup (WT) show increased replicative lifespan when exposed to chronic oxidative damage (*sod1Δ*–/+ Dsup (WT); *n* = 30 individuals for each background). Source data are provided as a Source Data file.

yielded no protection, with growth indistinguishable from an Empty-vector control strain. As such, the entire C-terminus of Dsup (aa 360-445) is important for protecting yeast from oxidative DNA damage.

Free-radical scavengers are effective at protecting yeast from oxidative stress and extending lifespan[35], so we investigated if Dsup mediates such a role. Redox-sensitive GFPs (roGFPs) are excited at 405 nm in oxidizing but 488 nm in reducing conditions, so relative emissions after excitation at [405/488 nm] report on changes in redox state. In this manner, H$_2$O$_2$ treatment of yeast cells caused the redox

state to increase in the cytoplasm and nucleus (as determined with compartment specific reporters: respectively roGFP2-Grx1[36] or roGFP2-Grx1-NLS) (Supplementary Fig. 2 and Fig. 3c). Of note, Dsup (WT), Dsup HMGN 3 R/3E and Dsup ΔHMGN ΔC + NLS each reduced the initial and post-H$_2$O$_2$ treatment redox state in the cytoplasm and nucleus (all relative to EV; Empty vector) (Fig. 3c). Importantly, these data show that while sequences within the Dsup N-terminal region (aa 1-359) reduce free-radical levels, this is insufficient to confer enhanced survival in response to H$_2$O$_2$ since Dsup ΔHMGN ΔC + NLS cells were not resistant to this genotoxin (Fig. 3a, b).

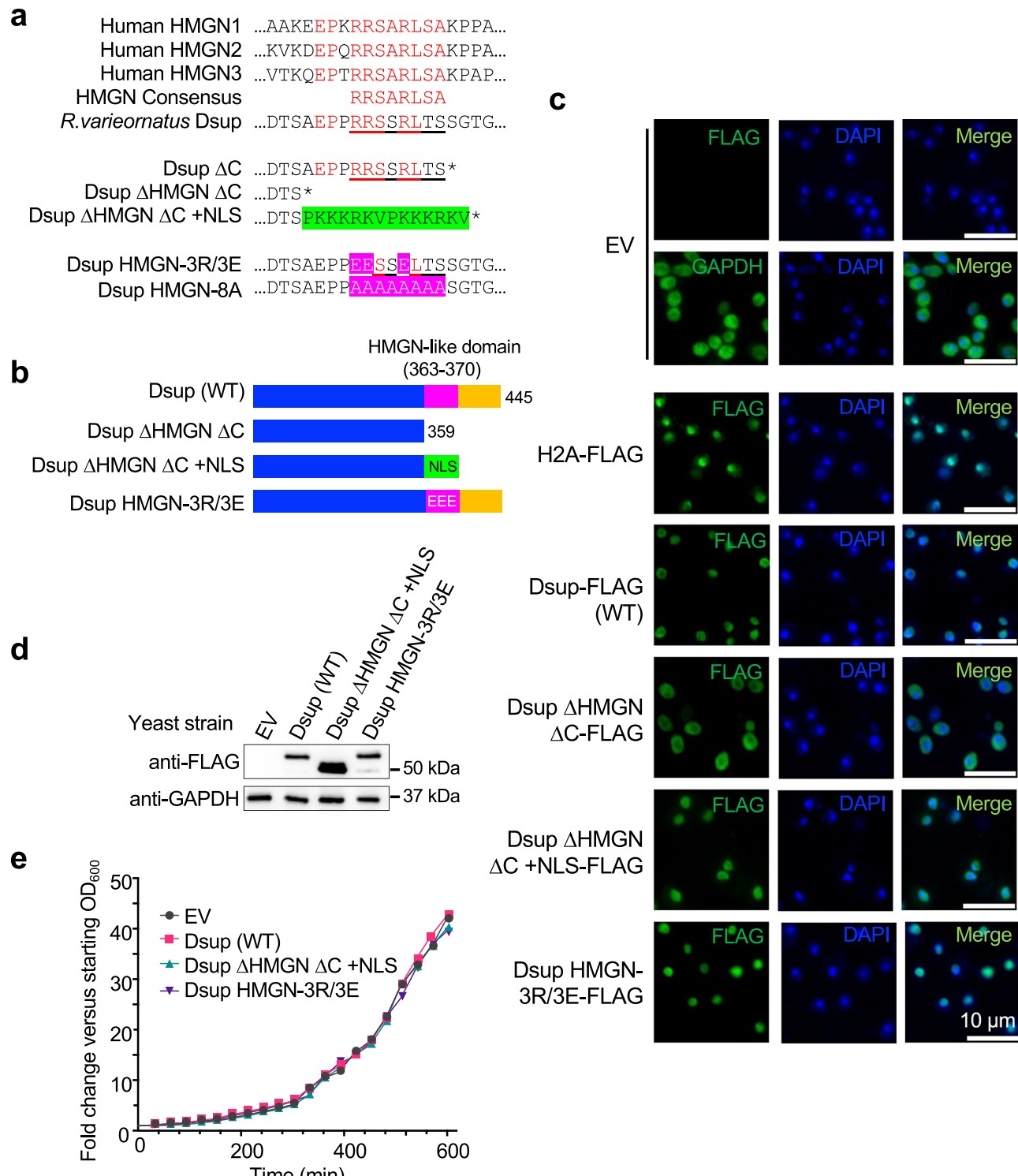

**Fig. 2 | Heterologous Dsup is nuclear localized in yeast and does not negatively impact growth. a** Alignment of human HMGN1-3 identifies the HMGN core consensus (RRSARLSA). Also shown the *R.varieornatus* Dsup HMGN-like motif (aa 363-370, RRSSRLTS [underlined]) and alleles that mutate or delete this area and/or the downstream C-terminal region (aa 371-445) for phenotypic and/or biochemical studies (adapted from[22]) (Supplementary Data File 1E). Residues in red, including three functionally important arginines, are identical between the HMGN core consensus and Dsup. Green indicates the duplicated SV40 nuclear localization signal (NLS: PKKKRKVPKKKRKV) added to Dsup ΔHMGN ΔC for yeast expression. Pink indicates mutated residues in the HMGN-like motif to create -3R/3E (charge

reversal) or -8A (charge neutralization). *, stop codon. **b** Schematic of Dsup wild-type (WT) and mutant alleles stably expressed in yeast for phenotypic studies (Figs. 1–4 [Dsup ΔC + NLS (as used in Fig. 7) not depicted]). N-terminal 6xHIS and C-terminal FLAG tags on each protein are not depicted. **c** Immunofluorescence to examine subcellular location of Dsup alleles (anti-FLAG) in yeast. DNA stain DAPI identifies nuclei. H2A-FLAG and GAPDH are respective controls for nuclear and cytoplasmic localization. EV, Empty vector. **d** Immunoblot shows relative expression of Dsup alleles (anti-FLAG) in yeast. Anti-GAPDH is a loading control from samples run on the same gel. **e** Representative growth curves of yeast expressing Dsup alleles. Source data are provided as a Source Data file.

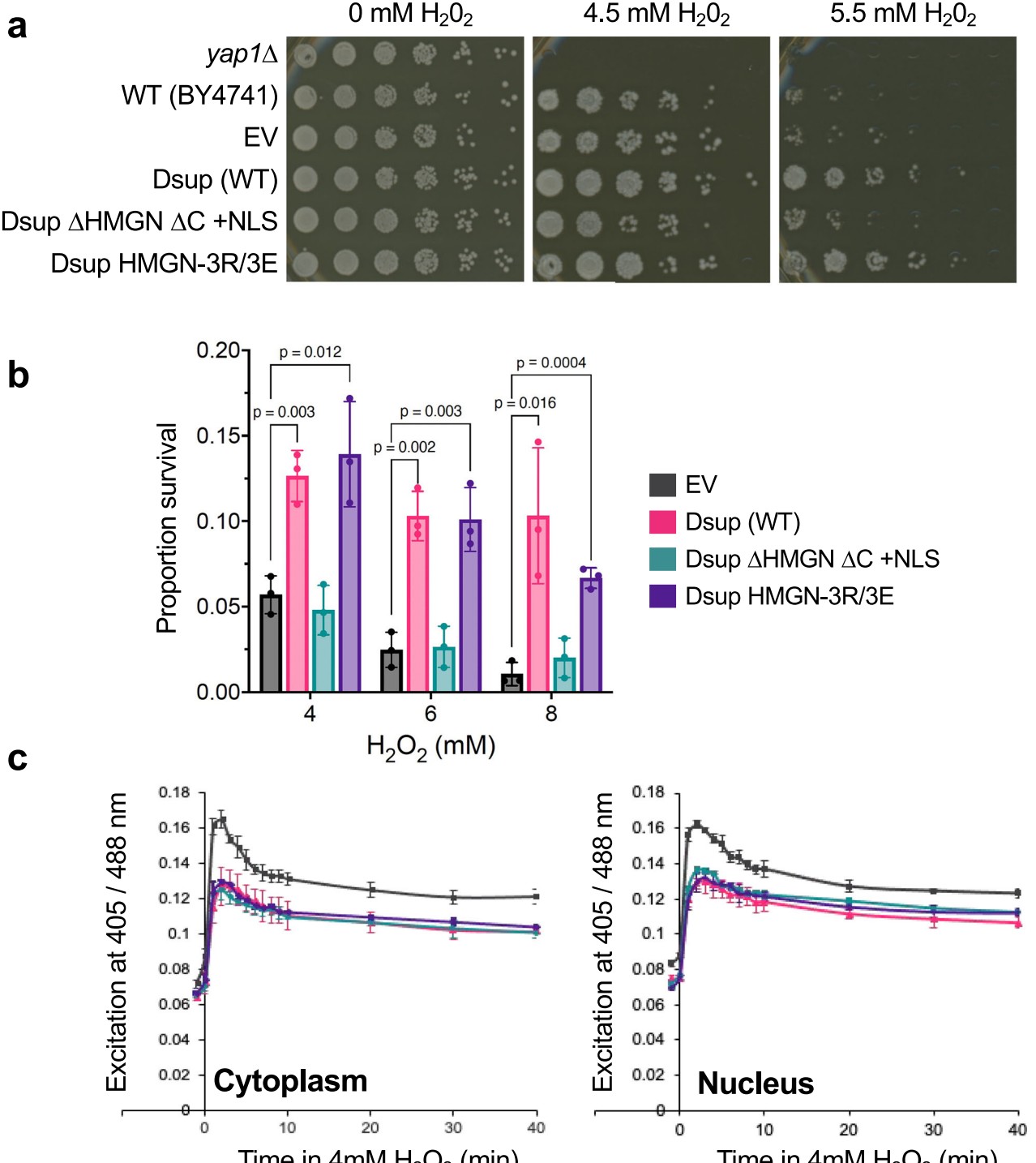

**Fig. 3 | Dsup promotes survival after oxidative DNA damage in a manner that requires its C-terminus but is not due to ROS scavenging. a** The Dsup HMGN-like motif (aa 363-370) and C-terminal region (aa 371-445) are required to confer yeast resistance to chronic $H_2O_2$-mediated oxidative damage (concentrations indicated). $yap1\Delta$ is a positive control for sensitivity to $H_2O_2$. WT, wild-type. EV, Empty vector. **b** The Dsup HMGN-like motif and C-terminal region are required to confer yeast resistance to acute $H_2O_2$-mediated oxidative damage. Graph plots percentage cell survival after 90-minute exposure to $H_2O_2$ (concentrations indicated). Shown are average and standard deviation of experiments using three independent colonies for each strain. *p*-values were determined using two-sided Student's t-test. **c** Relative redox state in the cytoplasm (left; roGFP2-Grx1 [redox sensitive GFP - GlutaRedoXin 1] reporter) and nucleus (right; roGFP2-Grx1-NLS reporter) for indicated yeast strains (color key as in **b**) through $H_2O_2$ exposure (4 mM; time in min). T0 sample was taken immediately after $H_2O_2$ addition. Shown are average and standard deviation of experiments performed from three independent colonies. Source data are provided as a Source Data file.

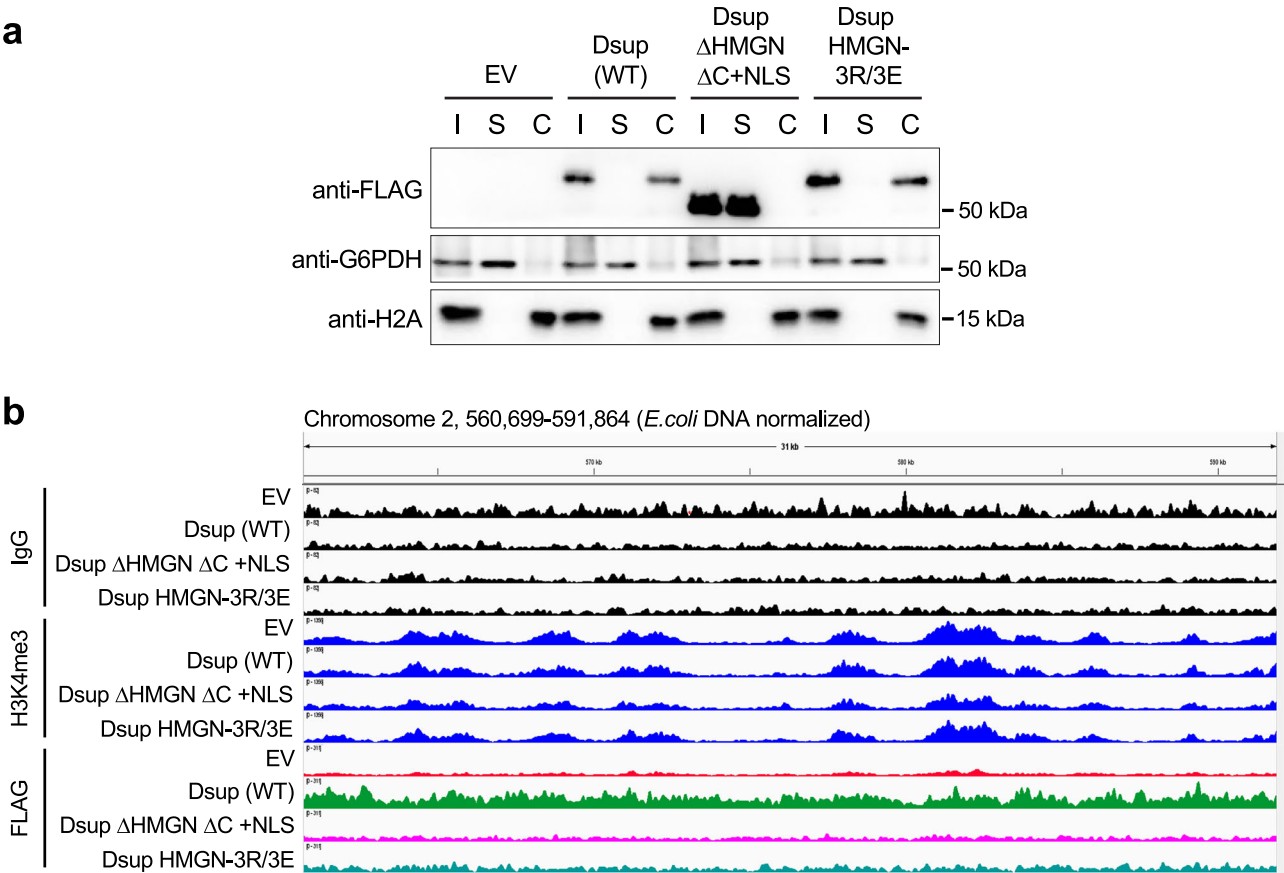

**Fig. 4 | Dsup fractionates with yeast chromatin and associates across the yeast genome without apparent bias. a** The Dsup HMGN-like motif (aa 363-370) and C-terminal region (aa 371-445) are required for the association with yeast chromatin. Yeast spheroblasts (Input) from indicated strains were resolved to Soluble and Chromatin fractions and immunoblotted as indicated. Confirming effective fractionation: GAPDH is a soluble protein, H2A is chromatin bound. EV, Empty vector. The samples detected for FLAG were resolved on one gel and the same samples were ran on a second gel to detect GAPDH and H2A. **b** CUT&RUN to examine Dsup allele interactions across the yeast genome (anti-FLAG). For each strain IgG (assay background) and anti-H3K4me3 (active gene promoters) were respectively included as negative and positive controls. Each target is group-scaled (after normalization to *E.coli* spike-in) to the highest signal in the depicted IGV window (IgG (82); H3K4me3 (1356) or Dsup-FLAG (311)). Data is from a representative CUT&RUN of biological replicates (sequence statistics in Supplementary Data File 2). Source data are provided as a Source Data file.

## Dsup binds chromatin throughout the yeast genome, in a manner dependent on sequences within the C-terminus

Dsup was first isolated from the chromatin fraction of Tardigrade cells[4], and shown to bind preferentially to nucleosomes over free DNA in vitro[22]. Therefore, we investigated if Dsup binds yeast chromatin in vivo. After cellular fractionation to separate chromatin-bound from soluble proteins, Dsup (WT) and Dsup HMGN-3R/3E both enriched in the chromatin fraction (Fig. 4a). By contrast, Dsup ΔHMGN ΔC + NLS was entirely soluble (Fig. 4a), suggesting that despite nuclear localization (Fig. 2c), it does not bind chromatin. Of note, the chromatin association of Dsup (WT) and Dsup HMGN-3R/3E, but not Dsup ΔHMGN ΔC + NLS, paralleled their ability to promote cell survival in the face of oxidative damage (Fig. 3a, b), suggesting that chromatin binding is key.

The presence of tardigrade Dsup in human and plant cells alters transcription factor binding and gene expression in response to DNA damage[5,10]. This is suggestive that Dsup binds certain areas of the genome to influence transcription. Alternatively, to have the greatest physically protective effect from oxidative DNA damage, Dsup might be expected to uniformly coat chromatin. To investigate these possibilities, we used Cleavage Under Targets & Release Using Nuclease (CUT&RUN)[37] to map 6His-Dsup-FLAG localization (by anti-FLAG), and observed that Dsup (WT) enriched across the yeast genome, without

bias or selectivity (i.e., not showing gaps, peaks or domains; Fig. 4b). Of note, the ability of CUT&RUN to map transcriptionally active promoters with anti-H3K4me3 was unaffected by Dsup (compare Empty vector (EV) and Dsup (WT)), indicating minimal impact on local chromatin structure (Fig. 4b). This was confirmed by micrococcal nuclease (MNase) digestion of chromatin from yeast −/+ Dsup, where we observed only minor differences in efficiency (Supplementary Fig. 3).

We next compared CUT&RUN across Dsup alleles, first noting that relative DNA yields post MNase digestion (prior to adapter ligation) were consistently Dsup (WT) » Dsup HMGN-3R/3E > Dsup ΔHMGN ΔC + NLS > EV) (Supplementary Fig. 4). This is mirrored in the CUT&RUN data, where Dsup HMGN-3R/3E showed less enrichment than Dsup (WT) across all genomic regions, while Dsup ΔHMGN ΔC + NLS resembled Empty vector (Fig. 4b; all data group scaled after normalizing to *E. coli* spike-in to allow comparisons of global changes in factor binding).

Taken together, these data indicate that Dsup binds without obvious bias across the genome in a manner dependent on its C-terminus (which includes the HMGN-like motif). Further, while mutation of conserved arginines in the Dsup HMGN-like motif (-3R/3E) reduced chromatin binding and/or increased its turnover (revealed by the long incubation steps of CUT&RUN but not direct chromatin fractionation (Fig. 4a)), the allele still conferred protection from

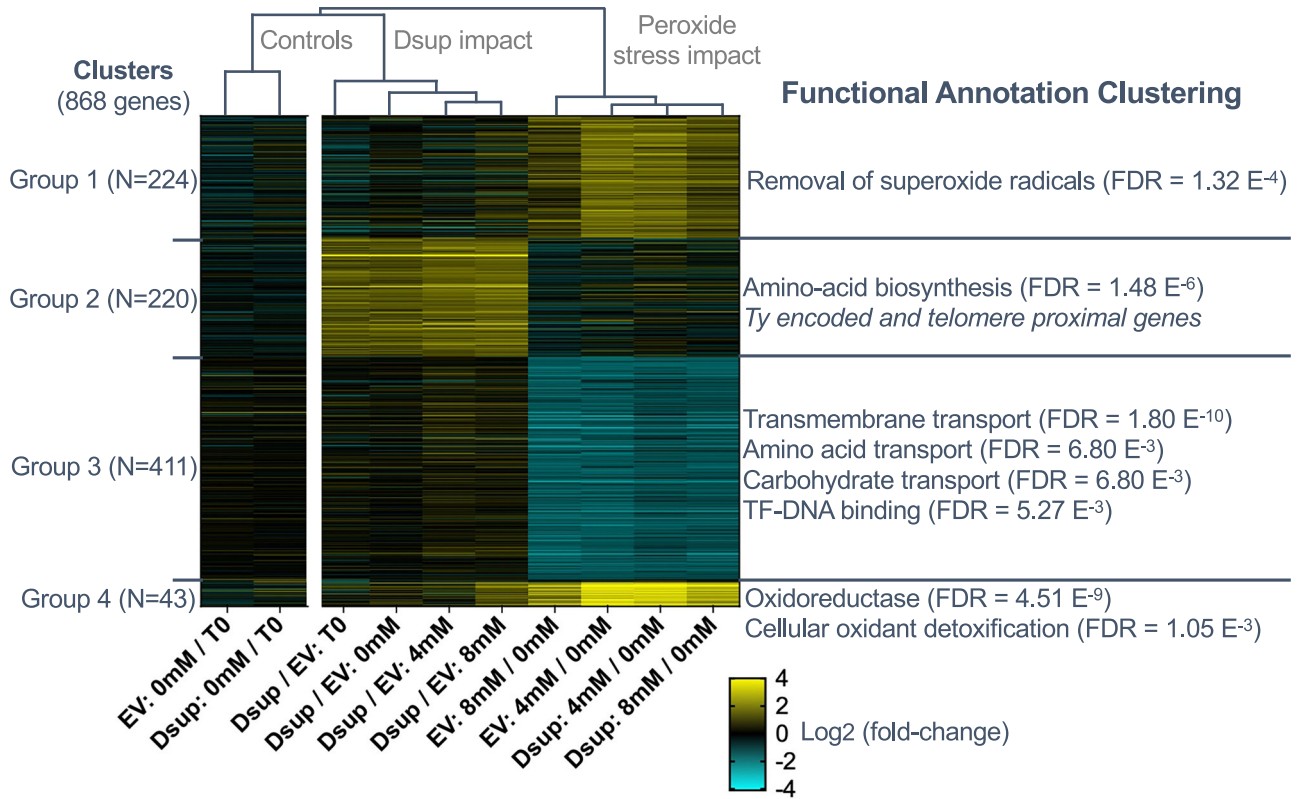

**Fig. 5 | Dsup expressing yeast do not show a poised or elevated transcriptional response to oxidative stress.** Yeast +/− Dsup were exposed to $H_2O_2$ (0, 4 or 8 mM for 30 min; T0 is untreated) and RNA-seq performed. Indicated comparisons show differentially expressed genes (Supplementary Data File 3) from the most significantly enriched functional annotation categories and their false discovery rates (FDR; as determined by DAVID). Group 2 Dsup-upregulated genes included a set involved in amino acid biosynthesis. However, this did not extend to activation of the 'General Amino Acid Response', including elevated protein levels of Gcn4 transcriptional activator[80] (Supplementary Fig. 5). Data are the average of three independent repeats. EV, Empty vector. Source data are provided as a Source Data file.

oxidative DNA damage (Fig. 3a, b). These data indicate that both the HMGN-like motif and distal C-terminal sequences contribute to global chromatin binding.

### Cells expressing Dsup are not transcriptionally primed for DNA repair nor do they have an enhanced transcriptional response to oxidative damage

Given that Dsup binds to yeast chromatin (Fig. 4), and this engagement is important for protection from $H_2O_2$ (Fig. 3), we next asked if Dsup also alters transcription in a manner that promotes the response to oxidative stress or DNA repair. Within the well characterized "environmental stress response" (ESR), genes involved in RNA processing and protein synthesis are transcriptionally repressed, while those involved in ROS detoxification and maintaining redox balance are activated[38–40]. To determine any Dsup influence on yeast gene expression we performed RNA-seq analysis. Here we observed transcriptional changes characteristic of the ESR 30 min after exposure of yeast cells −/+ Dsup to $H_2O_2$ (4 mM or 8 mM: Fig. 5 and Supplementary Data File 3). This included induction of genes involved in the removal of superoxide radicals (Group 1; 224 genes), and detoxification of cellular oxidants and oxidoreductases (Group 4; 43 genes). We also observed the characteristic transcriptional repression of transcription factors and proteins involved in amino acid and carbohydrate transport and transmembrane transport; halting these metabolic activities until the stress has been removed (Group 3; 220 genes). Notably, we did not observe an elevated ESR in cells expressing Dsup −/+$H_2O_2$ exposure (compare Dsup (WT) to Empty vector (EV) control: Fig. 5 and Supplementary Data File 3).

Of interest, Dsup expression per se led to transcriptional changes for 220 genes relative to Empty vector control (Group 2: Fig. 5 and Supplementary Data File 3). This included a subset of genes involved in amino acid biosynthesis, though it did not extend to increased abundance of Gcn4 transcriptional activator (and so did not represent a 'General Amino Acid Response'; Supplementary Fig. 5). However, Group 2 also included genes proximal to telomeres, or encoded from Ty transposable elements. Both are usually actively repressed locations in budding yeast[41], suggesting Dsup expression may lead to some opening of repressive chromatin; in line with the minor increase in MNase digestion efficiency also observed in these cells (Supplementary Fig. 3).

Together, this suggests that cells expressing Dsup are not transcriptionally poised to respond to oxidative stress, nor do they have an enhanced transcriptional response, including of DNA damage response genes, when it is encountered (Fig. 5). As such transcriptional profiling provides no explanation for the ability of Dsup to promote yeast cell survival in the face of oxidative damage.

### The Dsup HMGN-like motif mediates interactions with histones while the adjacent C-terminal region binds DNA, with each needed to fully protect yeast from oxidative damage

Having ruled out a transcriptional role for Dsup in protecting yeast from oxidative damage (Fig. 5), we set out to further characterize its interaction with chromatin (Fig. 4). To interrogate potential mode(s) of engagement, we used the Captify in vitro chemiluminescent assay[42]. Here the biotinylated target (e.g., free DNA or fully defined mononucleosome: Supplementary Data File 1D) couples to streptavidin-donor beads while epitope-tagged Query (e.g., various forms of 6His-

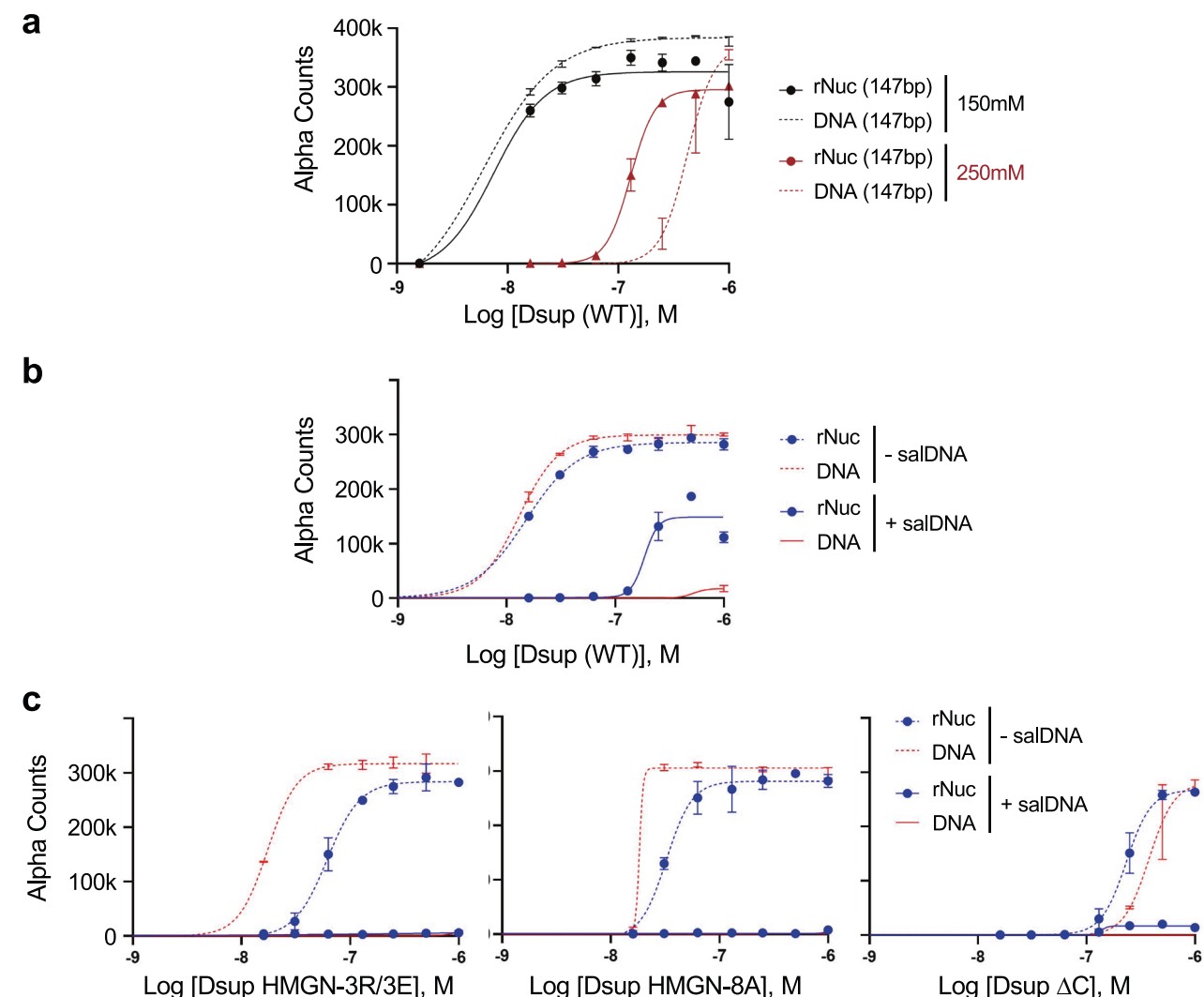

**Fig. 6 | The Dsup HMGN-like motif and C-terminal region each contribute to nucleosome binding. a** Impact of ionic strength (150 mM (normal saline, 0.9%) or 250 mM NaCl) on binding of Dsup (WT: 1 - 0 μM in two-fold serial dilutions) to unmodified nucleosome (rNuc on 147 bp DNA; 10 nM) or free DNA (147 × 601: 2.5 nM). **b** salDNA (1μg/ml in 150 mM NaCl) was a more effective competitor of Dsup binding to free-DNA *vs.* to nucleosomes, suggesting the latter involves multivalent engagement. **c** Mutation of the Dsup HMGN-like motif (aa 363-370; charge reversing -3R/3E, or charge neutralizing -8A) or deletion of the adjacent C-terminal region (Δ371-455; ΔC) each compromised DNA / nucleosome binding compared to Dsup (WT) (see also Supplementary Fig. 8). Note the relative binding of each allele to free DNA or nucleosome targets, where Dsup HMGN mutants preferred free DNA while Dsup ΔC preferred nucleosomes. Key and reaction conditions as in (**b**). Data are the average and standard deviation of two technical repeats. Source data are provided as a Source Data file.

Dsup-FLAG; Fig. 2a, Supplementary Fig. 6 and Supplementary Data File 1E[22]) couples to anti-tag acceptor beads. After mixing potential reactants the donor beads are excited at 680 nm, releasing a singlet oxygen that causes emission (520–620 nm) in proximal acceptor beads: this luminescent signal is directly proportional to the amount of [Donor - Acceptor] bridged by the [Target: Query] interaction (Supplementary Fig. 7). Binding is quantified by plotting Alpha Counts (fluorescence) as a function of protein concentration and expressed as relative EC50 (for all $EC_{50}^{rel}$ from this study see Supplementary Data File 4).

We first titrated salt (NaCl) to examine the potential impact of non-specific ionic interactions. At 150 mM Dsup (WT) similarly bound unmodified nucleosomes (rNuc) and naked DNA, but at 250 mM showed a distinct preference for nucleosomes (Fig. 6a). Choosing the ionic strength closest to normal saline (150 mM; ~0.9% NaCl), we next tested the impact of adding salmon sperm DNA (salDNA) competitor[42,43]. This identified an optimized condition where nucleosome binding was retained over free-DNA (Fig. 6b and Supplementary Data File 4), confirming that the Dsup association with DNA is a significant, but not exclusive, element of its interaction with chromatin[22].

We next used Captify to examine the contribution of various Dsup regions for nucleosome / DNA engagement. Here Dsup ΔHMGN ΔC showed no binding (Supplementary Fig. 8); not unexpected given our findings from cell fractionation and CUT&RUN (Fig. 4 and Supplementary Fig. 4). We thus created mutants to isolate the contributions of the HMGN-like motif (Dsup HMGN -3R/3E and Dsup HMGN-8A) or adjacent C-terminal region (Dsup ΔC) (Fig. 2a and Supplementary Data File 1E). Relative to Dsup (WT), each mutant had reduced binding to free DNA and nucleosomes that was undetectable in the presence of salDNA (compare Figs. 6b and 6c). Interestingly Dsup HMGN-3R/3E and -8A each preferred free DNA while Dsup ΔC preferred nucleosomes, indicating the mutated regions differentially contribute to chromatin engagement. Of note, while both HMGN mutants showed the same binding profile, charge reversing -3R/E bound nucleosomes -two-fold weaker than charge neutralizing -8A ($EC_{50}^{rel}$ 61.53 *vs.* 32.47 nM: Supplementary Data File 4), as might be expected given residue charge is central to how the HMGN-like motif engages the nucleosome acidic patch[32].

This data indicates the Dsup C-terminal domain (aa 208-445) previously defined as responsible for chromatin binding[4] actually

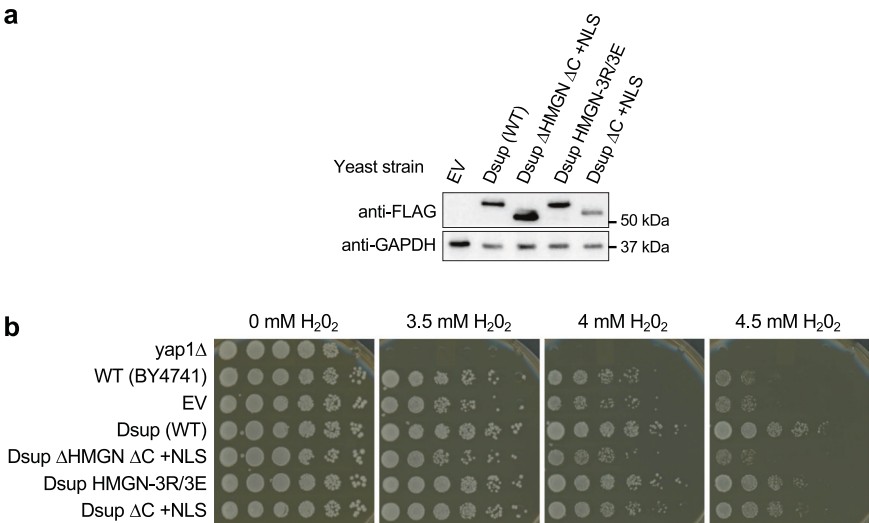

**Fig. 7 | The Dsup HMGN-like motif and C-terminal region each contribute to yeast survival during oxidative damage. a** Immunoblot shows relative expression of Dsup alleles (anti-FLAG) that delete or mutate (-3R/3E) the HMGN-like motif (aa 363-370) and/or C-terminal region (aa 372-455) in yeast (each N-terminal 6xHIS and C-terminal FLAG tagged; see also Fig. 2a–d). Anti-GAPDH is a loading control resolved on the same gel. EV, Empty vector. Dsup (WT), (aa 1-455). **b** Relative sensitivity of yeast strains (five-fold serial dilutions) to oxidative DNA damage by $H_2O_2$ (see also Figs. 1b and 3a). yap1Δ is a positive control for sensitivity to $H_2O_2$. WT (BY4741), wild-type yeast. Source data are provided as a Source Data file.

contains at least two functional elements: the nucleosome binding HMGN-like motif (aa 363-370) and DNA binding C-terminal region (aa 371-445). Our yeast analyses (Figs. 1–4) had not explored any specific contribution of the C-terminal region so we created a new deletion mutant (Dsup ΔC + NLS: Δ371-445 + NLS) for phenotypic testing. This was expressed in yeast at reduced levels relative to other forms of Dsup (Fig. 7a) but conferred some resistance to chronic $H_2O_2$ exposure (Fig. 7b: compare Dsup ΔC + NLS to BY4741 or EV). Thus, an intact HMGN-like motif or C-terminal sequences are each sufficient to grant nucleosome binding and protection from oxidative damage (albeit each weaker than Dsup (WT)), but loss of both regions yields a non-chromatin binding and non-protective Dsup allele.

## Dsup binds nucleosomes via multivalent interactions with the histone tails, acidic patch, and DNA

We next employed Captify with conditions optimized for multivalent engagement (150 mM NaCl, 1 μg/ml salDNA) and a diversity of fully defined nucleosomes (Supplementary Data File 1D) to determine which surfaces are engaged by Dsup. Here (and relative to rNuc) Dsup showed ever-decreasing binding to a H3 tail delete (H3.1 NΔ32 or H3.3 NΔ32), H4 tail delete (H4NΔ15), or 'tail-less' nucleosome (trypsin digested of all histone tails) (Fig. 8a). Supporting that Dsup may have a particular preference for the H4 tail, and further that this might be charge-state mediated (where acetylation neutralizes lysine charge), binding was reduced to H4tetraAc relative to H2AtetraAc or H3tetraAc nucleosomes (Fig. 8b). As predicted by CUT&RUN (Fig. 4b), [Dsup: Nucleosome] binding was not impacted by H3K4me3, nor by other lysine tri-methyl states (at H3K9, H3K27, H3K36 or H4K20: Fig. 8c). Finally mutations (H2AE61A, H2AE92K and H2BE105A/E113A) within the acidic patch, a hub of interaction for nucleosome binding proteins[44,45], abolished [Dsup: nucleosome] binding (Fig. 8d). Together, this indicates that the Dsup interaction with chromatin is mediated by DNA, the histone N-terminal tails, and the H2A/H2B nucleosome acidic patch (Fig. 9).

## Discussion

To explore the molecular basis of the extreme radioresistance of tardigrades, we investigated how *R.varieornatus* Dsup protects against oxidative damage in vivo. When expressed in budding yeast, Dsup coats the genome without apparent bias. This reduced oxidative DNA damage in a manner independent of ROS scavenging or amplifying transcriptional responses that protect from oxidative stress / enhance DNA repair. Rather, our data supports a model where Dsup uses at least two C-terminal regions to mediate multivalent interaction with several nucleosome surfaces and protect the underlying genome from oxidative DNA damage. Functionally this promotes yeast survival and longevity after exposure to elevated levels of hydroxyl radicals (Fig. 9).

Dsup is intrinsically disordered[17], which may allow it to dynamically wrap multiple nucleosome surfaces to shield DNA from damage. This includes a C-terminus enriched in positively charged amino acids to facilitate ionic interactions with negatively charged DNA[17] (Fig. 6), and an HMGN-like motif[22] for potential nucleosome binding. HMGN proteins were originally defined in vertebrates[46], where human HMGN2 uses its eponymous motif (RRSARLSA) to bind nucleosomes at the H2A/H2B acidic patch[32,47]. Dsup also binds the acidic patch (Fig. 8d), with this interaction lost in mutants that reverse or neutralize positively charged arginines in its HMGN-like motif (respectively Dsup HMGN-3R/3E or -8A: Fig. 6c). Individual disruption of the Dsup HMGN-like motif (HMGN-3R/3E) or adjacent C-terminal region (ΔC), yields mutants that retain some chromatin binding and improve yeast survival on exposure to oxidative damage (Figs. 3, 6 and 7). However, Dsup mutants lacking both regions (ΔHMGN ΔC) show no nucleosome / DNA binding and no protection (Supplementary Fig. 8 and Fig. 3). Redundancy in the multiple interactions between Dsup and chromatin would facilitate engagement even when certain surfaces are otherwise occupied by endogenous binding proteins, potentiating its ability to coat the genome. Binding studies with fully defined nucleosomes containing histone PTMs, tail deletions or acidic patch mutants show each surface contributes to Dsup engagement, with most PTMs of minimal impact (Fig. 8). Dsup could engage each of these surfaces independently or co-operatively: despite popular conception that the histone tails extend from the globular nucleosomal core[48], their default high affinity interaction is with nucleosomal DNA[49], which would co-localize at least two Dsup binding partners to promote engagement.

A recent cryo-EM [Dsup: nucleosome] structure supports a model where the HMGN-like motif effectively engages the acidic patch, while other Dsup regions transiently adopt multiple conformations around the nucleosome[19]: a binding mechanism in high concordance with this study. Further, we note *R. varieomatus* histones are highly conserved

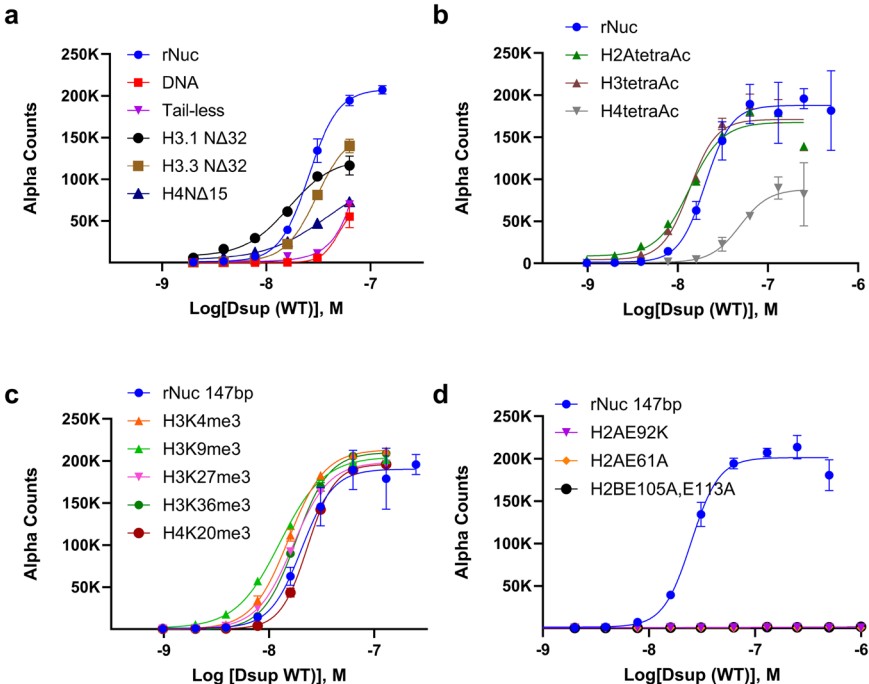

**Fig. 8 | Dsup binds nucleosomal DNA, histone tails and the acidic patch.** Relative interaction of Dsup (WT) with fully defined nucleosomes (Supplementary Data File 1d) containing histone tail truncations (**a**), lysine acetylations (**b**), lysine methylations (**c**), or acid patch mutations (**d**). All assays performed under optimized conditions (from Fig. 6b): Dsup (WT: 1 - 0 µM in two-fold serial dilutions), nucleosome (each as indicated; 10 nM), free DNA (147 × 601; 2.5 nM), 150 mM NaCl, 1 mg/ml salDNA. Data are the average and standard deviation of two technical repeats. Source data are provided as a Source Data file.

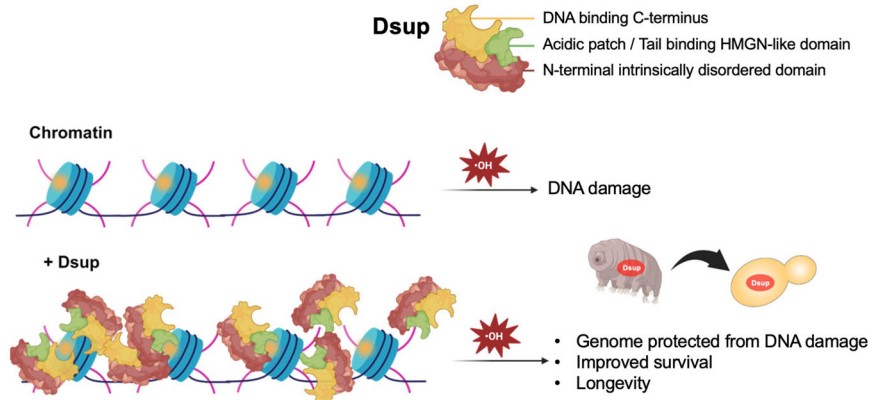

**Fig. 9 | Model for multivalent association of Dsup with the genome to protect from oxidative DNA damage.** Dsup chromatin-interacting regions include the histone tails / acid-patch binding HMGN-like motif (aa 363-370) and DNA-binding C-terminus (aa 372-455). Multivalent binding of tardigrade Dsup to the chromatinized yeast genome protects against oxidative DNA damage (as induced by $H_2O_2$ or elevated when *SOD1* (Superoxide dismutase 1) is deleted (e.g., Figs. 1, 3 and 7).

Independent mutations of the Dsup HMGN-like motif or C-terminal region weaken its association with nucleosomes, but these alleles still protect from oxidative DNA damage (e.g., Figs. 3, 4, 6 and 7). However simultaneous deletion of both regions (as in Dsup ΔHMGN ΔC or Dsup ΔHMGN ΔC + NLS) eliminates chromatin binding and genome protection (e.g., Fig. 4 and Supplementary Figs. 4 and 8). Created in BioRender. Khan, L.F. (2025) https://BioRender.com/16wiptg.

with those from human (as used in Captify) and yeast (with which Dsup associates in vivo) (Supplementary Data File 1E). We thus propose that the interactions dissected in this study represent the endogenous situation in tardigrades. Successful gene editing of *R.varieornatus* has recently been reported[50] so it is now possible to test this directly.

There is precedent for proteins binding the genome to protect from irradiation and $H_2O_2$. Chromatin compaction protects DNA from free radical-mediated damage caused by ionizing radiation or iron in vivo and in vitro[51–53], suggesting a direct mechanism rather than a particular feature of the cellular environment. Compacted chromatin also protects from ROS damage after direct incubation with $H_2O_2$[54].

Additionally, deleting proteins involved in chromatin assembly and disassembly, including the ISWI, Chd1, and INO80 remodelers, renders chromatin more sensitive to DNA damage[55]. Dsup can now be added to the list of proteins that protect the DNA from $H_2O_2$-induced damage by direct chromatin binding, though the precise mechanism remains to be determined.

It is notable that Dsup protected yeast from oxidative damage, but increased their sensitivity to MMS, bleomycin or UV (Fig. 1b). Meanwhile, Dsup confers protection from UV in humans and plants[5,10,56]. The reason for this difference is unknown, but it is possible that the stoichiometry of Dsup to nucleosomes matters (see below). Future studies

should examine if Dsup expression in yeast delays the repair of DNA lesions generated by various genotoxins, potentially due to hindering access of their repair machineries. We note, however, that the growth rate of Dsup expressing yeast was not impacted (Fig. 2d), indicating they are fully competent for transcriptional regulation, DNA replication and mitosis – events one could imagine might be compromised on coating the genome with a heterologous protein. That this is tolerated is almost certainly due to [Dsup: nucleosome] complexation being highly dynamic[19] (Fig. 4).

Dsup expression in other species induces transcriptional changes that could grant improved DNA repair and thus contribute to their genotoxin resistance[5,10,56]. This includes a heightened transcriptional response to oxidative stress in rice[57], and induced expression of DNA repair genes after UV or $H_2O_2$ exposure in human cells[10]. However, flies expressing Dsup show a decreased transcriptional response to irradiation, despite their increased resistance and fewer DNA double-strand breaks[6]. We found no evidence that Dsup expressing yeast primed or hyperactivated the known transcriptional responses that protect from environmental (including oxidative) stress or enhance DNA repair (Fig. 5). The transcriptional effects of Dsup expression thus appear quite distinct in different systems. Dsup expression in flies downregulated 733 genes (with only two upregulated), including those involved in transcription, chromatin assembly and DNA replication, suggesting the protein acts as a non-specific repressor[6]. By contrast, Dsup expression in yeast led to increased expression of 220 genes, most of which were normally repressed transposons or telomere-proximal genes (Fig. 5) suggesting that chromatin at these regions was made more accessible to the transcription machinery. How this could occur warrants further investigation.

As noted above, it is possible that species-specific effects of Dsup expression are due to the expression levels in different organisms, suggesting titration is needed to receive beneficial effects. Indeed, in initial testing we expressed Dsup from yeast promoters of various strengths, but only that of highly expressed *TDH3*[58] protected from oxidative damage (Fig. 1b and not shown). This yielded a Dsup expression level equivalent to that of histone H2B (Fig. 1a), suggesting the availability of at least two molecules per nucleosome. Given this Dsup expression level protects the genome from oxidative DNA damage (Fig. 1c), is bound to chromatin genome-wide (Fig. 4), and redundantly interacts with multiple nucleosome surfaces (Figs. 6 and 8), it likely non-specifically coats the in vivo genome to physically protect from oxidative damage, as proposed from in vitro studies[22]. It may be relevant to note that when we yeast-codon optimized *R.varieornatus* Dsup in an attempt to promote still higher expression levels, the resulting yeast were inviable, suggesting that too much Dsup is deleterious. In agreement, the expression of codon-optimized Dsup had detrimental effects in cultured rat neurons[59].

Despite not having an enhanced environmental stress response (ESR), Dsup expressing yeast had a lowered redox state (Fig. 3). However, this does not explain their resistance to oxidative stress, since non-protective Dsup ΔHMGN ΔC + NLS also effectively lowered redox potential (Fig. 3). This suggests sequences within the N-terminal portion of Dsup (aa 1-359) reduce the redox state, though the mode of action is unclear. Dsup does not include any cysteine residues so it is unlikely to directly act as an antioxidant by donating hydrogens from thiol groups[60]. The effect could be indirect, since Dsup expression in tobacco pollen leads to increased levels of antioxidants, polyphenol and flavonoids[61]. Future studies should thus examine if the ability of the Dsup N-terminus to reduce the redox state in yeast is via increased levels of endogenous antioxidants.

It is intriguing that the Dsup protein appears to have evolved multiple independent mechanisms to enable tardigrades to withstand oxidative damage. This is transferrable by heterologous expression to other species and appears to manifest by a mix of reduced redox state, transcriptional changes that enhance repair, and multivalent chromatin binding for physical protection. Our studies in budding yeast suggest the last is of greatest relevance but this may be due to tightly controlled expression that achieves a stoichiometry comparable to core-histones. The effective therapeutic harnessing of Dsup will require careful optimization of its expression in whichever context it is employed.

# Methods

## Yeast strains, primers and plasmids

Plasmid pRS306-PTDH3-Dsup was created by Gibson cloning of amplified DNA fragments following kit directions (NEB Gibson Assembly® Cloning Kit). In brief, pRS306[62] was digested with SacI and BglII. The *TDH3* promoter was PCR amplified (primers in Supplementary Data File 1A) from yeast genomic DNA with pTDH3_SacI_F (giving homology to Sac1 digested end of pRS306) and pTDH3_R (giving homology to 5' end of Dsup gene). The *R. varieornatus* Dsup gene (aa1-445; encoding protein accession P0DOW4, Supplementary Data File 1E) including N-terminal 6xHis and C-terminal FLAG epitope tags was amplified from plasmid pET21b-nHis6-Rvar-DSUP-cFLAG (kind gift from James Kadonaga[22]), with primers Rvar_Dsup_F and Rvar_Dsup_R, respectively giving homology to the *TDH3* promoter 3' end and *ADH1* terminator 5' end. The *ADH1* terminator was amplified from yeast genomic DNA using primers tADH1_F and tADH1_BglII_R, respectively giving homology to the Dsup gene 3' end and BglII digested pRS306 5' end.

pRS306-PTDH3-Dsup was digested with MfeI and integrated to BY4741[63] at the endogenous *TDH3* promoter to make yeast strain RGY002 (pTDH3-6His_Dsup_FLAG: all yeast strains and their phenotypes in Supplementary Data File 1B). Mutations to integrated Dsup, including insertion of a stop codon to derive the 'Empty vector' strain (RAY149), were by CRISPR-Cas9 mediated genome editing[64]. Primer sequences to generate guide RNAs[61] and HDR template DNA are in Supplementary Data File 1A. Yeast were handled using standard methods. Dsup expressing strains were grown in SC-ura media (unless otherwise indicated).

P415TEF cyto roGFP2-Grx1-NLS was made from p415TEF cyto roGFP2-Grx1 (kind gift from Tobias Dick; Addgene plasmid # 65004)[36] by traditional cloning. First, the roGFP2-Grx1 sequence was PCR-amplified from plasmid p415TEF cyto roGFP2-Grx1 using primers that added a 2xNLS sequence (PKKKRKVPKKKRKV) to the Grx1 C-terminus. The resulting PCR product was digested with BamHI and HindIII and ligated to similarly digested plasmid p415TEF cyto roGFP2-Grx1 (Supplementary Data File 1B). All yeast strains and plasmids are available on request.

## Immunoblot analysis

Exponentially growing yeast cells ( ~ $10^7$ at $OD_{\lambda600}$ 0.8–1.0) were collected by centrifugation, washed once with water, and flash frozen in liquid nitrogen. Pellets were resuspended in 100 µL modified Laemmli buffer[65], boiled for five minutes (mins), and clarified by centrifugation. Proteins in the supernatant were resolved by 10% SDS-PAGE, membrane transferred, and immunoblotted with antibodies to FLAG (Sigma F1804; 1:1,000), GAPDH (Sigma A9521; 1:10,000), H2B (Abcam ab1790; 1:5000), H2A (Active Motif 39235; 1:2000), GCN4 (Absolute Antibody Ab00436-1.1; 1:1000), or H3 (Abcam ab1791; 1:1000). Uncropped unprocessed blots are provided in the Source Data.

## ELISA for 8-OhdG

30 mL yeast cultures were grown at 30 °C in shaking flasks to $OD_{\lambda600}$ 0.6. Cells were harvested by centrifugation, and half of each culture resuspended in 15 mL of fresh SC-ura media −/+10 mM $H_2O_2$. After two hours at 30 °C cells were harvested by centrifugation, and genomic DNA isolated (Thermo Scientific Yeast DNA extraction kit) and resuspended in 50 µL nuclease-free water. DNA was quantified (NanoDrop spectrometer), samples

diluted to 2 mg/mL, and sequentially incubated at 95 °C for five mins and on ice for 10 min. 50 µg DNA was then sequentially incubated with nuclease P1 (NEB; 1 U at 37 °C for two hours in provided buffer), alkaline phosphatase (NEB Quick CIP; 10 U at 37 °C for one hour in provided buffer supplemented with 100 mM Tris pH 8), at 95 °C for 10 min, and centrifuged at 6000 x g for five mins. Samples were then DNA quantified (NanoDrop spectrometer) and normalized.

ELISA to measure 8-hydroxy 2-deoxyguanosine (8-OhdG) was performed as per kit instructions (Abcam ab201734). For each sample 15 µg DNA was assayed in triplicate (three independent cultures for each condition) and absorbance at 450 nm measured by plate reader.

### Growth curve analysis
Yeast cultures were grown to saturation overnight in YPD at 30 °C and diluted to $OD_{\lambda 600}$ 0.1–0.2. The $OD_{\lambda 600}$ of cultures from three independent colonies per genotype were measured every 30 min, and growth curves fitted with an exponential regression in Microsoft Excel. Doubling times were calculated as the slope of the curve during exponential phase and compared across independent cultures using a Student's t-test.

### Immunofluorescence analysis
Indirect immunofluorescence of yeast cells was carried out as previously[66]. 2.5 OD of early-mid log phase cells ($OD_{\lambda 600}$ 0.5–0.6) were crosslinked in 4% formaldehyde for 20 min at room temperature, then spheroplasted with 500 µg/mL Zymolyase 100 T for 30 min at 30 °C with rotation. Spheroplasted cells were applied to a 10-chamber poly-lysine coated microscope slide and permeabilized by incubation in methanol at −20 °C for six min, immediately followed by incubation in acetone at −20 °C for 30 seconds. After blocking in 5% BSA, slides were incubated with anti-FLAG (Sigma F1804; 1:1,000), anti-H2A (Abcam ab18255; 1:1.000) or anti-GAPDH (Sigma A9521; 1:5,000). After washing, slides were incubated with Alexa Fluor® 594 or 488 secondary antibodies (BioLegend), and coverslips mounted using ProLong™ Gold Antifade Mountant with DAPI (Invitrogen). Images were taken using an Olympus BX63 Fluorescence Microscope with a DP80 Camera and 60X objective.

### Acute and chronic damage sensitivity analysis
To measure the response to acute hydrogen peroxide ($H_2O_2$) exposure, yeast cells were grown in YPD media to mid-log ($OD_{\lambda 600}$ 0.5–1.0), harvested by centrifugation, and resuspended to $OD_{\lambda 600}$ 0.6 in fresh media containing $H_2O_2$ (0, 4, 6, or 8 mM). After 90 min incubation (30 °C with shaking), cultures were washed and serial dilutions spread on SC-ura agar plates. After two days at 30 °C, colonies were counted and averaged across three technical replicates. Three independent experiments were performed from separate starting colonies, and statistical analyses performed using a Student's t-test.

To measure the response to chronic $H_2O_2$ exposure, yeast cells were grown in liquid culture until mid-log, harvested by centrifugation, and resuspended in sterile water to $OD_{\lambda 600}$ 1.0. Five-fold serial dilutions were made in a 96-well plate and spotted with a sterile 6x8-prong manifold onto YPD agar plates containing indicated concentrations of $H_2O_2$. Similar methods were used to evaluate sensitivity to methyl methanesulfonate (MMS; at indicated concentrations in YPD) and Zeocin (at indicated concentrations in YPD). To measure sensitivity to ultraviolet light, yeast serial dilutions on YPD plates were exposed to UV (doses (J/cm²) indicated in figures) using a crosslinker (Stratalinker). Plates were incubated for three days at 30 °C.

### Replicative lifespan analysis
Yeast cells were grown overnight to early-mid log ($OD_{\lambda 600}$ 0.2–0.6) and diluted to OD 0.1 in freshly-filtered YPD. This inoculum was added

to an automated dissection chip (iBiochips) as per manufacturer's instructions to achieve single cell loading. Light microscopy images of cells were acquired every 20 min over four days using an Evos FL Auto two-cell imaging microscope and associated software (ImageJ). At least 50 cells were counted per condition, with survival curves calculated in Graphpad Prism 9, and statistical analyses performed with a log-rank test.

### Chronological lifespan analysis
Yeast chronological lifespan was measured as previously[67]. Data is presented as average and standard deviation across three independent cultures (each an average of two technical replicates).

### Redox analysis
Yeast cells expressing cytoplasmic or nuclear localized (Supplementary Fig. 2) roGFP (redox sensing GFP) from respective plasmids p415TEF cyto roGFP2-Grx1 (GlutaRedoXin 1)[36] or roGFP2-Grx1-NLS were grown in SC-LEU media to mid-log ($OD_{\lambda 600}$ 0.6–0.8) and diluted to $OD_{\lambda 600}$ 0.6 in 5 mL flow cytometry tubes. Fluorescence at 405 nm and 488 nm was measured on a flow cytometer (BD Biosciences BD® LSR II) immediately before addition of $H_2O_2$ (to 4 mM), and at indicated time points. Data is presented as the mean and standard deviation of 405/488 nm values for each timepoint (independent triplicates) using FlowJo.

### Chromatin fractionation analysis
Exponentially growing yeast cells ($\sim 4 \times 10^8$ at $OD_{\lambda 600}$ 0.8–1.0) were collected by centrifugation, washed once with ice cold 10% glycerol, and flash frozen. After thawing on ice, the cell pellet was washed (100 mM Tris pH 9.4, 10 mM DTT), resuspended in the same buffer, and rested on ice for 10 min. Cells were pelleted by centrifugation, washed in spheroplasting buffer (10 mM HEPES, 1.2 M Sorbitol, 0.5 mM PMSF), resuspended in spheroplasting buffer containing 56 µg/mL Zymolyase 100 T (US Biological), and incubated at 30 °C for one hour with rotation (until a 50 µL aliquot mixed with 1 mL 10% SDS had an $OD_{\lambda 600}$ ~10% of the starting value). Spheroplasts were collected by centrifugation (1500 g for 2 min) and sequentially washed with spheroplasting buffer and wash buffer (1 M sorbitol, 20 mM Tris pH 7.5, 20 mM KCL, 2 mM EDTA, 0.5 mM PMSF, 0.1 µM spermine, 0.25 µM spermidine, 1:100 Calbiochem Protease Inhibitor Cocktail Set IV). Cells were gently resuspended and lysed in 250 µL Lysis Buffer (wash buffer with 400 mM sorbitol) for 10 min on ice.

Half of the volume after lysis (Input) was mixed with 5x Laemmli buffer, while the other half was pelleted at 14,000 x g for 15 min. The supernatant was collected (non-chromatin fraction) and mixed with 5x Laemmli buffer, and the pellet (chromatin fraction) resuspended in 1x Laemmli buffer. All fractions were boiled for 5 min, clarified by centrifugation, 7.5% of the total volume resolved by 12.5% SDS-PAGE, membrane transferred, and immunoblotted with anti-FLAG (Sigma F1804; 1:1,000) to detect Dsup. Successful fractionation was confirmed with anti-H2A (Abcam ab18255; 1:5,000) as a chromatin bound protein, and anti-GAPDH (Sigma A9521; 1:20,000) as a non-chromatin bound protein.

### CUT&RUN analysis
Nuclei from yeast cells expressing Dsup alleles (Supplementary Data File 1B) were purified as previously[68] with minor modifications. In brief, yeast cells were grown to mid-log ($OD_{\lambda 600}$ 0.6–0.8) in 500 mL SC-ura and collected by centrifugation. Cells were resuspended to 500 µL, spheroplasted with Zymolyase 100 T (2 mg/mL; 37 °C for 30 min), nuclei isolated as previously[68], and 1 mL aliquots ($5 \times 10^7$ nuclei) slow-frozen overnight in an isopropanol chamber at −80 °C.

For CUT&RUN[69], nuclei were rapidly thawed (2–3 min at 37 °C), and 100 µL of suspension used per reaction with the CUTANA™ ChIC/

CUT&RUN Kit (*EpiCypher*). In brief, after immunotethering (of pAG-MNase to Rabbit IgG, anti-H3K4me3, or anti-FLAG: Supplementary Data File 1C), MNase digestion was performed (4 °C for 2 h) and DNA eluted in 12 μL final volume. 5 ng of DNA was used to prepare sequencing libraries with the Ultra II DNA Library Prep Kit (*NEB* #E7645L). Libraries were sequenced on an Illumina NextSeq 2000 platform, obtaining an average of ~1.1 million paired-end (PE) reads per reaction (Supplementary Data File 2).

PE fastq files were aligned to the *sacCer3* reference genome (Bowtie2[70]), filtered from duplicate (SAMtools[71]), multi-aligned (broadinstitute.github.io/picard), and exclusion list reads[72], and the resulting unique reads for comparable reactions normalized by a spike-in scaling factor (1/ % *E.coli* reads) (BEDTools v2.30.0[73]) to further RPKM (Reads Per Kilobase per Million mapped reads) -normalized bigwig files (DeepTools). Peak visualization from bigwig files was by Integrative Genomics Viewer (IGV). CUT&RUN sequence data is available from NCBI Gene Expression Omnibus (accession number GSE237436). The CUT&RUN analyses were performed independently three times with consistent results.

## RNA-seq analysis

Overnight cultures of yeast strains −/+Dsup (RAY149 and RGY002 respectively: Supplementary Data File 1B) were grown at 30 °C in SC-URA medium. Cultures were diluted to $OD_{\lambda600}$ 0.2 in 30 ml SC-URA media, grown to mid-log ($OD_{\lambda600}$ 0.6–0.8), standardized to $OD_{\lambda600}$ 0.5 and split to 12 × 2 mL aliquots. An untreated triplicate (T0) was collected by centrifugation, washed in water and flash frozen in liquid nitrogen. To the remaining aliquots in triplicate $H_2O_2$ (Fisher Scientific BP2633-500) was added (0, 4, or 8 mM) and cultures incubated at 30 °C for 30 min. Cells were collected by centrifugation, washed in water, and flash frozen.

RNA extraction was performed with the MasterPure Yeast RNA Purification Kit (Epicentre MPY03010) using the standard protocol including DNA removal step. RNA quality, concentration and purity was determined by Tapestation (Agilent 5067-5576) and nanodrop (Fisher Scientific), and material stored at −80 °C until library preparation. For each biological replicate, three technical replicates of non-strand specific libraries were prepared from 200 ng mRNA (Fast RNA-seq Lib Prep Kit v2; Abclonal RK20306) and sequenced to an average depth of 25 million paired-end 150 x 150 bp. Library preparation and sequencing services were provided by Novogene.

Sequencing read and base-calling quality were assessed by FastQC v0.11.9 (www.bioinformatics.babraham.ac.uk/projects/fastqc/), and paired-end fastq files aligned to the *sacCer3* reference genome (Bowtie2[70]). Files were triple filtered from duplicate (SAMtools[71]), multi-aligned (broadinstitute.github.io/picard), and exclusion list reads[72] before further analysis. Gene FPKM (Fragment Per gene-length Kilobase per Million filtered reads) were computed with Homer v4.11[74]. Technical and biological replicate FPKM reproducibility were assessed by linear regression R-squared values. To improve the statistical power of differential gene expression analyses, technical and biological replicates were combined by EdgeR v3.34.0[75]. Genes were significantly differentially expressed if their false discovery rate was <0.05 and the fold-change was greater or less than two ($log_2$ <−1 or >1).

To generate the heat map in Fig. 5 the union of 868 significant genes from the ten pairwise $log_2$ ratios was clustered along each row (gene) by K-means and hierarchically by each column (pairwise ratios) using Cluster 3.0[76]. The optimal number of clusters were heuristically determined by the lowest K-means value that resulted in non-redundant cluster patterns. To functionally annotate the clusters, gene set enrichment analysis (GSEA) was performed on each of the five clusters with DAVID Knowledgebase v2024q2[77,78] (all data in Supplementary Data File 3). RNA-sequence data is available from NCBI Gene Expression Omnibus (accession number GSE294109).

## Microccocal nuclease (MNase) analysis of yeast chromatin

Yeast cultures were grown to mid-log phase and crosslinked using 1% (w/v) formaldehyde in either minimal media or water. Cell pellets equivalent to 50 ml of 0.85 OD culture were collected, resuspended in 1 ml lysis buffer (50 mM HEPES-KOH pH 8.0, 150 mM NaCl, 2 mM EDTA, 1% (v/v) Triton X-100, 0.1% w/v sodium deoxycholate), mixed with 1 ml zirconia beads and lysed using a Mini-Bead beater. The resulting extracts were spun in a bench-top centrifuge (16,000 g for 10 min at 4 °C) and supernatants discarded. Chromatin-containing pellets were resuspended, washed once with NP-S buffer (0.5 mM Spermidine, 0.075% (v/v) IGEPAL, 50 mM NaCl, 10 mM Tris-Cl pH 7.5, 5 mM $MgCl_2$, 1 mM $CaCl_2$) and resuspended in NP-S buffer + 1 mM β-mercaptoethanol (β-ME). 5 ml cell equivalent of each sample was removed as Input. To parallel 5 ml cell equivalents 20U of MNase (Worthington Biochemical corporation) was added, and reactions incubated at 37 °C for 15 or 20 min. Digestions were halted by EDTA (to 10 mM) and incubating on ice for 10 min, following by centrifugation (16,000 g for 10 min at 4 °C) to collect the supernatant containing digested DNA fragments. Crosslinks were removed from Input and MNase digested samples by resuspending in NP-S buffer + 1 mM β-ME, mixing with an equivalent volume of 2x proteinase K buffer (40 mM Tris-Cl pH 7.5, 40 mM EDTA, 2% (w/v) SDS), adding 3 μl of 20 mg/ml proteinase K, and incubating at 65 °C overnight. DNA was purified from each sample using phenol-chloroform-isoamyl alcohol extraction and isopropanol precipitation. Material (and DNA size standards) were resolved on a 1.3% agarose gel and stained with ethidium bromide before image capture (Supplementary Fig. 3).

## PTM-defined nucleosomes

All mononucleosomes (*EpiCypher*; Supplementary Data File 1D) were created from fully-defined (PTM or mutant) octamers wrapped by 5′ biotinylated 147 × 601 DNA (Supplementary Data File 1E) unless otherwise stated, with PTMs confirmed by mass-spectrometry and immunoblotting (if an antibody was available)[42,79]. Histone tail truncations were by direct expression of the indicated histone prior to octamer assembly (H3.1 NΔ2, H3.1 NΔ32, H3.3 NΔ32 or H4 NΔ15), or trypsin digestion of an assembled unmodified nucleosome (tail-less).

## Recombinant Dsup proteins

Dsup alleles for recombinant protein expression were cloned (GenScript) into pET-21a expression vectors with N-terminal 6x Histidine and C-terminal FLAG epitope tags (Supplementary Data File 1E). For expression, individual colonies from freshly transformed *E.coli* BL21(DE3) (T7 Express *lysY/I*⁻ for HMGN-8A) were grown to saturation overnight and used to inoculate 1 L LB supplemented with 100 μg/mL carbenicillin. Cultures were grown to $OD_{\lambda600}$ ~ 0.7 (200 rpm shaking at 37 °C), induced with 0.5 mM IPTG (200 rpm shaking at 16 °C overnight), and harvested by centrifugation (4000 g for 20 min at 4 °C). Pellets were resuspended in 12.5 mL wash buffer (50 mM Tris pH 7.5, 500 mM NaCl) per liter of culture and transferred to 50 mL conical tubes for centrifugation (4000 g for 15 min at 4 °C). Supernatants were decanted and cell pellets frozen at −80 °C.

Dsup (WT) and HMGN 3 R/3E were purified essentially as previously[22]: In brief, cell pellets from a 2 L culture were thawed, resuspended in 25 mL lysis buffer (50 mM Tris pH 7.5, 500 mM NaCl, 1 mM PMSF, 0.2 mg/mL lysozyme, 0.2% Triton X-100, and Roche protease inhibitor tablet (one per 50 mL lysis buffer)), and incubated at 4 °C for 1 h with nutating. Lysates were sonicated (20 sec pulse on, 40 sec pulse off, 30% amplitude, 3 min total: *Qsonica* Q500) and clarified by centrifugation (31,000 g for 20 min at 4 °C). Lysate soluble fraction was mixed with 15 mL nickel resin (HisPur™ Ni-NTA: *Thermo Scientific*) equilibrated in cold Buffer A (50 mM Tris pH 7.5, 500 mM NaCl, 5 mM imidazole) and batch bound at 4 °C for 30−60 min with nutating. Subsequent steps of column purification were performed using a gravity column at room temperature with ice cold buffers.

Unbound material was collected as flow-through, the column washed with 5 column volumes (CV) Buffer A, and eluted with 4 CV Buffer B (50 mM Tris pH 7.5, 500 mM NaCl, 500 mM imidazole), collecting one fraction per CV. Fractions E1 and E2 were combined, treated with 1 μl Benzonase (*Millipore Sigma*) on ice for ~ 15 min, mixed with 1 mL Pierce™ Anti-DYKDDDDK Affinity Resin (*Thermo Scientific*) equilibrated in Buffer C (50 mM Tris pH 7.5, 150 mM NaCl, 1 mM EDTA, 10% glycerol), and incubated at 4 °C for 1 hour with nutating. Subsequent steps of column purification were performed using a gravity column at room temperature with ice cold buffers. Unbound material was collected as flow-through, and the resin washed with 5 CV Buffer C and eluted with 10 CV Buffer D (50 mM Tris pH 7.5, 150 mM NaCl, 1 mM EDTA, 10% glycerol, 0.3 mg/mL 3X FLAG peptide). Aliquots were analyzed by SDS-PAGE / Coomassie staining, and fractions containing Dsup proteins combined in SnakeSkin dialysis tubing and dialyzed at 4 °C (three 2 L buffer exchanges (overnight, 2 hrs, 2 hrs) of 50 mM Tris pH 7.5, 150 mM NaCl, 20% glycerol, 1 mM DTT). Proteins were collected and additional buffer exchanges (to concentrate and remove the FLAG peptide) were by centrifugation (Vivaspin 20 (*Cytiva*) 10 K MWCO centrifugal filter). Aliquots were dispensed (0.1 mL), flash frozen in a dry ice / 100% ethanol bath, and stored at −80 °C.

Dsup ΔC and HMGN-8A were purified as above with minor changes: Cell pellets from a 2 L culture were lysed and clarified, and the soluble fraction decanted to a small beaker. 10% polyethyleneimine (branched, ~M.N. 60,000, 50 wt % aqueous solution pH 7.6; *Thermo Scientific*) was added dropwise to a final concentration of 0.1% while stirring at 4 °C. After continued stirring for ~ 45 min, lysate was clarified by centrifugation (31,000 g for 20 min at 4 °C) and soluble material decanted to a new tube for subsequent purification. Proteins were acid-eluted from the anti-DYKDDDDK Affinity Resin with 5 CV buffer E (0.1 M glycine pH 2.8) and immediately neutralized with 0.5 M Tris pH 8.5. Positive fractions were pooled for further purification.

### Captify binding assays

The assay previously known as dCypher™ is now named Captify™, with no distinction in how the assay is performed or its capabilities. The interaction of WT or mutant 6His-Dsup-FLAG (kind gift from James Kadonaga[22]: *aka*. the Queries [Supplementary Data File 1E]) with free DNA (147 × 601 Widom sequence) or fully defined nucleosomes (the Targets: Supplementary Data File 1D) was assayed by Captify on the Alpha (Amplified luminescence proximity homogeneous assay) platform as previously[42,79] with minor modifications.

Dsup queries (5 μL) were serially titrated in duplicate against a fixed concentration of target (5 μL, 10 nM biotinylated nucleosome or 2.5 nM free DNA (147 × 601)) in 384-well plates and incubated for 30 min. A 10 μL mix of AlphaScreen streptavidin Donor (Revvity, 6760002) and nickel-chelate Acceptor beads (Revvity, AL108M) was added and incubated for a further 60 min. All incubations were at room temperature in subdued lighting. Alpha counts were measured using a PerkinElmer 2104 EnVision plate reader (680 nm laser excitation, 570 nm emission filter ± 50 nm bandwidth).

[Query: Target] binding was examined over a range of assay conditions (20 mM Tris pH 7.5, 0.01% BSA, 0.01% NP-40, 1 mM DTT with additives as noted), including the impact of ionic strength (50−250 mM NaCl) and competitor salmon sperm DNA (salDNA; 0−20 μg/mL). Binding curves [Query: Target] were generated using a non-linear 4PL curve fit in Prism 9.0 (GraphPad) to yield $EC_{50}^{rel}$ values[42,79] (Supplementary Data File 4). Where necessary, values beyond the Alpha hook point (indicating bead saturation / competition with unbound Query) were excluded and top signal constrained to average max signal for Target. In cases where signal never reached plateau, those were constrained to the average max signal within the assay (relative to unmodified nucleosome). In cases where a targets max signal never achieved half of max signal relative to unmodified

nucleosome, an $EC_{50}^{rel}$ was deemed not determinable (ND). In cases where a targets max signal never surpassed a two-fold increase, an $EC_{50}^{rel}$ was deemed ND at less than lowest concentration tested (e.g., ND; <1 nM).

### Statistics and reproducibility

Each experiment in this study was repeated independently three times with similar results. No data were excluded from the analyses. No statistical method was used to predetermine sample size. The investigators were not blinded to allocation during experiments and outcome assessment.

### Reporting summary

Further information on research design is available in the Nature Portfolio Reporting Summary linked to this article.

## Data availability

CUT&RUN sequence data is publicly available from NCBI Gene Expression Omnibus at accession number GSE237436. RNA-sequence data is publicly available from NCBI Gene Expression Omnibus at accession number GSE294109. Source data are provided with this paper.

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

## Acknowledgements

We are indebted to Drs. Jim Kadonaga, George Kassavetis, Grisel Cruz-Becerra, and Carolina Chavez for discussing pre-published work and generously sharing plasmids and purified proteins. We are grateful to Andres Mansisidor for help generating plasmids and yeast strains, and the Weill Cornell Flow Cytometry Core for technical support. *EpiCypher* is supported by National Institutes of Health (NIH) grants R44 HG010640, R44 GM117683, R44 GM136172 and R44 CA212733. CKC is supported by NIH training grant T32 GM135128 and BDS by NIH grant R35 GM126900. JKT is supported by NIH grants R35 GM139816 and R01 CA95641. RRA was supported by a Medical Scientist Training Program grant from the National Institute of General Medical Sciences of the NIH under award number T32 GM007739 to the Weill Cornell / Rockefeller / Sloan Kettering Tri-Institutional MD-PhD Program.

## Author contributions

R.A. constructed all strains, and performed western blotting, immunofluorescence, serial dilution, ROS and DNA damage analyses, with assistance of N.S., A.S., K.B., R.G. and K.K.; L.F.K. performed Captify and CUT&RUN assays under supervision of MRM. CKC performed RNA-seq studies under supervision of B.D.S.; N.A. performed replicative aging analyses under supervision of I.G.; S.D.L.P. performed chronological aging analyses. T.G. performed Redox State assays under supervision of I.G.; U.C. performed the MNase assays. A.R.H. and B.J.V. provided data analysis and interpretation for CUT&RUN and RNAseq. R.W., R.J.E. and HEW created fully defined histones, octamers, and/or nucleosomes. S.R.H. and L.M. provided recombinant Dsup proteins. M.W.C. was responsible for approach conception. M.C.K. and J.K.T. were responsible for program conception, project supervision and support in data analysis and interpretation. R.A. and J.K.T. drafted the original manuscript with support from L.F.K., B.D.S., M.R.M. and M.C.K.; All authors contributed to and are responsible for subsequent versions.

## Competing interests

EpiCypher is a commercial developer and supplier of reagents (*e.g.,* fully defined semi-synthetic nucleosomes (dNucs) and SNAP-CUTANA® K-MetStat spike-ins) and platforms (*e.g.,* Captify™ and CUTANA™ CUT&RUN) used in this study. LFK, ARH, RW, SRH, LM, MWC, MRM and MCK are currently employed by (and own shares in) EpiCypher. RJE and HEW were previously employed by (and own shares in) EpiCypher. MWC, BDS and MCK are board members of EpiCypher. The remaining authors declare no competing interests.
