## [Peer Review file · Nature Communications]

Multivalent binding of the tardigrade Dsup protein to chromatin promotes yeast survival and longevity upon exposure to oxidative damage

Corresponding Author: Dr Jessica Tyler

Version 0:

Reviewer comments:

Reviewer #1

(Remarks to the Author)

The tardigrade-specific Dsup protein is a chromatin-binding factor that protects human cells, flies, and tobacco plants against hydroxyl radical-mediated DNA damage. In the submitted work, the authors examined the effect of Dsup in the budding yeast *Saccharomyces cerevisiae* and analyzed its interaction with chromatin. Yeast cells containing Dsup show higher tolerance to hydrogen peroxide (which produce reactive oxygen species) relative to control cells that do not contain Dsup (Fig. 1). Dsup in yeast is localized to the nucleus if it contains a nuclear localization sequence (NLS, endogenous or from SV40 T ag), and the presence of Dsup at levels used in this study did not affect yeast growth rate (Fig. 2). The C-terminal region of Dsup (amino acids 360 – 445; see Dsup deltaC, Fig. 2a) was observed to be important for the survival of yeast to treatment with hydrogen peroxide (Figs. 3 and 7) as well as for the binding of Dsup to chromatin in yeast cells and in vitro (Figs. 4 and 6). The HMGN-like region, which was probed by using a 3R to 3E mutation (Dsup 3R/3E; Fig. 2a), was found to exhibit reduced binding to chromatin that is nevertheless sufficient for protection of yeast against hydrogen peroxide treatment (Figs. 3, 4, 6, 7). The extensive analysis of the binding of Dsup to various mutant or covalently-modified histones revealed that the H3 and H4 tails and the H2A and H2B acidic patch are important for Dsup binding to nucleosomes (Fig. 5). Mutant Dsup that contains the HMGN-like region but lacks the C-terminus (Dsup HMGN deltaC + NLS; Fig. 7a) was found to protect yeast cells from treatment with hydrogen peroxide (Fig. 7). Overall, these findings led to model proposed in Fig. 8.

This manuscript provides new and substantial data on the mechanism by which Dsup binds to chromatin and provides protection of living cells against damage by reactive oxygen species. Particular strengths of this paper are the analysis of Dsup in yeast cells in the presence versus the absence of reactive oxygen species and the detailed study of the binding of Dsup to differently modified nucleosomes. The manuscript was also very well written. There are some specific issues, which are listed below, that would be good for the authors to address. Overall, publication in Nature Communications is recommended.

Specific comments

1. The authors conclude that the C-terminal region of Dsup binds to DNA. It remains possible, however, that the mutant HMGN-like region (3R to 3E) can bind to DNA in the absence of sequences from 371 to the C-terminus (445). In other words, the authors have not shown that the C-terminal region (371-445) can bind to DNA in the absence of the HMGN-like region (363-370). Hence, if the authors wish to conclude that the C-terminal region alone (371-445) can bind to DNA, it would be necessary to test and to analyze something like a Dsup mutant that lacks the HMGN-like region (e.g., Dsup delta 363-370).
2. Fig. 5a is somewhat confusing. It would be helpful if it were redrawn. Some problems include: need to explain that delta 1 O₂ is singlet oxygen; should make it more clearly apparent that there is streptavidin on the donor bead; indicate that the green ball is biotin; the salDNA should have a closer resemblance to the DNA in the nucleosome; is that a tardigrade on top of the acceptor bead? if so, maybe it would be best to remove, as it unnecessarily complicates the figure.

Other comments

1. Page 9: in human cells, physiological salt is about 142 mM K⁺ and about 7-15 mM Na⁺. Thus, I would not say that 150

mM Na⁺ is "physiological" (unless that is physiological in yeast).

2. Fig. 1b. "cm²" should have the "2" as a superscript.

3. Fig. 4b. Did you perform replicates of the CUT&RUN experiments? If not, I would recommend that you do so.

4. Figs. 6a and 6b. It appears that some error bars might be missing.

Reviewer #2

(Remarks to the Author)

In this paper, the authors report the results of the study of the expression of *Ramazzottius varieornatus* Dsup in *Saccharomyces cerevisiae*. They show that in this model DSUP reduces oxidative DNA damage and extends the lifespan of budding yeast exposed to ROS, in a manner which requires either the Dsup HMGN-like domain or C-terminal sequences. The paper is overall clearly and well written. It reports, for the first time, the expression of DSUP in a yeast model and deepens the mechanisms responsible for Dsup-mediated protection after oxidative damage. The reference to updated literature is complete. The methodology sounds and the experiments are accurately described. Nevertheless, in my opinion, the manuscript has some limitations that need to be overcome to make it sharper and more complete.

1) First, the authors should define what slant to give the paper.

The results of the study mainly shed light on the role of Dsup in DNA protection after oxidative stress and on Dsup interaction with DNA. However, the authors state that the research can allow understanding of "the molecular basis of the extreme radioresistance of tardigrades", even though the experiments have been performed in an in vitro model that is not a Tardigrade. They also state that this work can "provide precedent for the development of organisms that can survive and live longer in the face of oxidative damage, potentially expanding the range of applications for developing therapeutic interventions by biotechnology, and furthering efforts towards human resistance to extraterrestrial effects". Which kind of organisms? Which therapeutic interventions and how can they increase human resistance to extraterrestrial effects?

2) In the budding yeast model, Dsup decreases survival in response to non-oxidative DNA-damaging agents, including UV. This is quite surprising, since an increased resistance to UV exposure have been shown in several systems, including human cells (PMID 37511223 and 34681069), *E. Coli* (PMID: 34970246) and plants (PMID: 32955680). How do the authors explain it? This point deserves to be deepened in the Discussion.

3) The experiments performed in the study show that in the yeast Dsup has no influence on the nuclear redox state and does not act as a scavenger. But what happens in other parts of the cell (cytoplasm and mitochondria)? An increasing of detoxification mechanisms aimed at removing ROS and limiting oxidative stress has been reported in HEK293 cells. Even though the authors show that Dsup uniformly coats the genome, a modulation of gene expression in response to ROS is still conceivable. This aspect has been completely neglected; however, it would be important for the thoroughness of her paper.

4) The CUT&RUN and MNase digestion methods should be more extensively described in the Materials and Methods section, since they are not very commonly used.

5) It would be better to move the description of the setup of dCypher protocol to the Materials and Methods

6) The sentence "The observed sensitivity of yeast expressing Dsup (WT) to MMS, Zeocin and UV (Fig. 1b) was not seen upon expression of Dsup 3R/3E or Dsup. C+NLS (not shown), indicating that while Dsup 3R/3E can protect from oxidative damage, it does not fully mimic the WT protein" (pag. 6) is not very clear.

7) Please, check that all the abbreviations are spelled-out the first time that are used in the text.

Version 1:

Reviewer comments:

Reviewer #1

(Remarks to the Author)

The revised paper looks good. I recommend acceptance as is.

Reviewer #2

(Remarks to the Author)

I have read with great interest the revised version of the paper by Aguilar and colleagues. The manuscript has been greatly improved and is now much more comprehensive and well structured.

I just have a few minor suggestions that I think could make the article even clearer.

1. Section "The Dsup C-terminus is required for protection of yeast from oxidative damage, in a manner independent of ROS scavenging": the authors should better clarify the concept of "in a manner independent of ROS scavenging", as in reality the presence of Dsup, even modified, reduces the redox state in both the cytoplasm and the nucleus. In addition, the observation that sequences within the Dsup N-terminal region (aa 1-359) reduce free radical levels, while this does not confer enhanced survival in response to H₂O₂ (since Dsup Δ HMGN Δ C +NLS cells were not resistant to this), is very relevant and should be more clearly stated.

2. Another part that needs further development is the section "Cells expressing Dsup are not transcriptionally primed for

DNA repair nor do they have an enhanced transcriptional response to oxidative damage". First of all, I would suggest moving this part either to the end of the Results or after the first section ("Heterologous expression of *R. varieornatus* Dsup in budding yeast protects against oxidative damage and promotes longevity in the face of increased oxidative stress") so as not to create breaks between the parts dealing with Dsup-DNA interaction at different levels. I also think that the authors should explain this part better. In Dsup+ there is no increased ESR compared to EV. However, Dsup expression led to transcriptional changes in response to H₂O₂ for 220 genes that were not similarly affected in EV. Thus, a difference at the transcriptional level is detectable and deserves attention for further investigation. This point should be made more explicit.

Version 2:

Reviewer comments:

Reviewer #2

(Remarks to the Author)

The authors have exhaustively replied to my comments. The manuscript is now suitable for publication.

Point-by-point response to reviewers' comments

R#1:

1. The authors conclude that the C-terminal region of Dsup binds to DNA. It remains possible, however, that the mutant HMGN-like region (3R to 3E) can bind to DNA in the absence of sequences from 371 to the C-terminus (445). In other words, the authors have not shown that the C-terminal region (371-445) can bind to DNA in the absence of the HMGN-like region (363-370). Hence, if the authors wish to conclude that the C-terminal region alone (371-445) can bind to DNA, it would be necessary to test and to analyze something like a Dsup mutant that lacks the HMGN-like region (e.g., Dsup delta 363-370).

Response: In response to this suggestion we made a new Dsup construct targeting the HMGN-like motif (aa 363-370, RRSARLSA). Of note we deliberately chose to replace the entire motif with charge-neutralizing alanines (Dsup HMGN-8A) rather than delete, and thus potentially disrupt Dsup structure. The resulting protein behaved similarly to charge-reversing Dsup-3R/3E: specifically compromised for nucleosome (vs. free-DNA) binding, and unable to bind the nucleosome in the presence of salmon sperm DNA (salDNA) competitor (**Fig. 6C**). We also made a new Dsup construct lacking the adjacent C-terminal region [Dsup Δ C], that is specifically compromised for free DNA binding (**Fig. 6C**). This data indicates the Dsup C-terminal domain (aa 208-445) previously defined as responsible for chromatin binding actually contains at least two functional elements: the nucleosome binding HMGN-like motif (aa 363-370) and DNA binding C-terminal region (aa 371-445).

2. Fig. 5a is somewhat confusing. It would be helpful if it were redrawn. Some problems include: need to explain that delta 1 O₂ is singlet oxygen; should make it more clearly apparent that there is streptavidin on the donor bead; indicate that the green ball is biotin; the salDNA should have a closer resemblance to the DNA in the nucleosome; is that a tardigrade on top of the acceptor bead? if so, maybe it would be best to remove, as it unnecessarily complicates the figure.

Response: We have made the suggested alterations and also moved this figure panel to the supplement (**Suppl. Fig. 7**) since numerous papers have used the methodology since our first submission.

Other comments

1. Page 9: in human cells, physiological salt is about 142 mM K⁺ and about 7-15 mM Na⁺. Thus, I would not say that 150 mM Na⁺ is "physiological" (unless that is physiological in yeast).

Response: As suggested, we have changed the language to: '*Choosing the ionic strength closest to normal saline (150 mM; ~0.9% NaCl), we next tested the impact of adding salmon sperm DNA (salDNA) competitor*'.

2. Fig. 1b. "cm²" should have the "2" as a superscript.

Response: We have now corrected this error, thank you.

3. Fig. 4b. Did you perform replicates of the CUT&RUN experiments? If not, I would recommend that you do so.

Response: The CUT&RUN experiments were performed three times with equivalent results (representative browser track data shown in **Fig. 4b** but see also **Suppl. Fig. 4**), and this is stated in the methods section.

4. Figs. 6a and 6b. It appears that some error bars might be missing.

Response: Where error bars appeared missing in the previous data, it was because they were too small to be obvious. However, during revision we uncovered contaminating DNA acting as a competitor in the original Dsup samples, and so repeated all dCypher analyses with new material. The revised data (triplicates with error bars: **Figs. 6** and **8**) is new and entirely consistent.

R#2:

1. First, the authors should define what slant to give the paper. The results of the study mainly shed light on the role of Dsup in DNA protection after oxidative stress and on Dsup interaction with DNA. However, the authors state that the research can allow understanding of “the molecular basis of the extreme radioresistance of tardigrades”, even though the experiments have been performed in an in vitro model that is not a Tardigrade.

Response: We appreciate this point. Given we are working with a tardigrade protein in a heterologous system we have now changed “*understand*” to “*providing insights into*” the molecular basis of their extreme radioresistance. As in the discussion (**p13**): ‘*Further, we note R.varieomatus histones are highly conserved with those from human (as used in dCypher) and yeast (with which Dsup associates in vivo) (Suppl. Table 1E). We thus propose that the interactions dissected in this study represent the endogenous situation in tardigrades. Successful gene editing of R.varieornatus has recently been reported so it is now possible to test this directly.*

They also state that this work can “provide precedent for the development of organisms that can survive and live longer in the face of oxidative damage, potentially expanding the range of applications for developing therapeutic interventions by biotechnology, and furthering efforts towards human resistance to extraterrestrial effects”. Which kind of organisms? Which therapeutic interventions and how can they increase human resistance to extraterrestrial effects?

Response: We have changed the emphasis of this element of the discussion. Dsup has now been expressed in a variety of heterologous systems and consistently transfers enhanced genotoxin resistance (so a highly active study topic): we compare the potential

mechanisms revealed from our comprehensive studies with what has been observed and proposed elsewhere.

2. In the budding yeast model, Dsup decreases survival in response to non-oxidative DNA-damaging agents, including UV. This is quite surprising, since an increased resistance to UV exposure have been shown in several systems, including human cells (PMID ¹ and ²), E. Coli (PMID: 34970246) and plants (PMID: 32955680). How do the authors explain it? This point deserves to be deepened in the Discussion.

Response: As requested, we have added a discussion section describing these observations and potential reasons for the differences. Noteworthy, the systems where expression of Dsup leads to UV resistance also showed changes in gene expression of repair proteins, while yeast showed no UV resistance and no changes in gene expression of repair proteins. This could be species-inherent or due to the relative Dsup expression level in each study.

3. The experiments performed in the study show that in the yeast Dsup has no influence on the nuclear redox state and does not act as a scavenger. But what happens in other parts of the cell (cytoplasm and mitochondria)? An increasing of detoxification mechanisms aimed at removing ROS and limiting oxidative stress has been reported in HEK293 cells. Even though the authors show that Dsup uniformly coats the genome, a modulation of gene expression in response to ROS is still conceivable. This aspect has been completely neglected; however, it would be important for the thoroughness of her paper.

Response: In response to the reviewer's comments, we have now performed extensive additional analyses:

- We found a reduced redox state in the nucleus and cytoplasm prior to / after H₂O₂ exposure in yeast cells expressing Dsup (**Fig. 3c**). Significantly, we see an identical redox response in cells expressing Dsup deleted for both the HMGN-like motif and C terminal sequences (Dsup Δ HMGN Δ C), even though these yeast are not resistant to H₂O₂ (**Figs. 3**). This suggests that Dsup does reduce the redox state via activities in the protein N terminus, but this is independent from the resistance to oxidative stress / chromatin binding activities that require the C terminus.
- An extensive RNA-seq analysis finds Dsup expression does not prime or hyperactivate the known transcriptional responses that protect from oxidative stress (here treatment with H₂O₂) or enhance DNA repair (**Fig. 5**).

We are grateful to the reviewer for their suggestion, which has improved the robustness of our analyses and insight to Dsup activities.

4. The CUT&RUN and MNase digestion methods should be more extensively described in the Materials and Methods section, since they are not very commonly used.

Response: We have added more detail to the **CUT&RUN method** and cited a recent protocol from our group (Firestone et al (2024) *bioRxiv* [10.1101/2024.12.03.626419](https://doi.org/10.1101/2024.12.03.626419)). We have also added a new section on the **MNase digestion method**.

5. It would be better to move the description of the setup of dCypher protocol to the Materials and Methods

Response: To aid the reader who might not be familiar with the approach, we have kept a brief description of dCypher in the results section. However there is a more extensive description in the methods section, and the approach schematic is now moved to supplemental material (**Suppl. Fig. 7**).

6. The sentence “The observed sensitivity of yeast expressing Dsup (WT) to MMS, Zeocin and UV (Fig. 1b) was not seen upon expression of Dsup 3R/3E or Dsup. C+NLS (not shown), indicating that while Dsup 3R/3E can protect from oxidative damage, it does not fully mimic the WT protein” (pag. 6) is not very clear.

Response: We have chosen to remove this sentence as it does not add to the study and discussed data that was not shown.

7. Please, check that all the abbreviations are spelled out the first time that are used in the text.

Response: We have carefully gone through the text and spelled-out additional abbreviations.

We truly thank the reviewers for their time, care and considered suggestions, as we feel that this is a more rigorous and compelling report as a result.

Point-by-point response to reviewer comments

Reviewer #1

Recommend acceptance as is [for which our thanks].

Reviewer #2

We thank the reviewer for their considered review and constructive suggestions.

1. Section "The Dsup C-terminus is required for protection of yeast from oxidative damage, in a manner independent of ROS scavenging": the authors should better clarify the concept of "in a manner independent of ROS scavenging", as in reality the presence of Dsup, even modified, reduces the redox state in both the cytoplasm and the nucleus. In addition, the observation that sequences within the Dsup N-terminal region (aa 1-359) reduce free radical levels, while this does not confer enhanced survival in response to H₂O₂ (since Dsup Δ HMGN Δ C +NLS cells were not resistant to this), is very relevant and should be more clearly stated.

Response: We have now more clearly stated that while Dsup (or more specifically N-terminal sequences) reduces redox potential in the yeast nucleus and cytoplasm, this does not promote survival upon oxidative damage (which is mediated by C-terminal sequences). Emphasizing this we changed the relevant section title to “**Dsup promotes yeast survival upon oxidative damage via its C-terminal sequences, and reduces the cellular redox state via its N-terminal sequences**” and made related statements in the results and discussion.

2. Another part that needs further development is the section "Cells expressing Dsup are not transcriptionally primed for DNA repair nor do they have an enhanced transcriptional response to oxidative damage". First of all, I would suggest moving this part either to the end of the Results or after the first section ("Heterologous expression of R. varieornatus Dsup in budding yeast protects against oxidative damage and promotes longevity in the face of increased oxidative stress") so as not to create breaks between the parts dealing with Dsup-DNA interaction at different levels. I also think that the authors should explain this part better. In Dsup+ there is no increased ESR compared to EV. However, Dsup expression led to transcriptional changes in response to H₂O₂ for 220 genes that were not similarly affected in EV. Thus, a difference at the transcriptional level is detectable and deserves attention for further investigation. This point should be made more explicit.

Response: We have more clearly stated the rationale for presenting the transcription data after showing that Dsup binds to chromatin in yeast: “**Given that Dsup binds to yeast chromatin (Fig. 4), and this engagement is important for protection from H₂O₂ (Fig. 3), we next asked if Dsup also alters transcription in a manner that promotes the response to oxidative stress or DNA repair.**” In terms of flow we consider it important to leave all the *in vivo* work together; rule out that improved survival after oxidative damage might be transcriptionally mediated; then transition to further *in vitro* analyses by stating “**Having ruled out a transcriptional role for Dsup in protecting yeast from oxidative damage (Fig. 5), we set out to further characterize its interaction with chromatin (Fig. 4). To interrogate potential mode(s) of engagement, we used the Captify *in vitro* chemiluminescent assay.**” Regarding the transcriptional changes in cells expressing Dsup, we thank the reviewer for making us realize that a text error persisted through multiple rounds of editing – a subset of transcriptional changes are directly induced by

Dsup independent of hydrogen peroxide treatment (Group 2, 220 genes in **Fig. 5**). We have corrected the text error / discussed the effect more clearly in the results and discussion sections.